# A transcomplementing gene drive provides a flexible platform for laboratory investigation and potential field deployment

Víctor López Del Amo [1], Alena L. Bishop [1], Héctor M. Sánchez C. [2], Jared B. Bennett[3], Xuechun Feng [1], John M. Marshall [2], Ethan Bier[1,4] & Valentino M. Gantz [1]*

CRISPR-based gene drives can spread through wild populations by biasing their own transmission above the 50% value predicted by Mendelian inheritance. These technologies offer population-engineering solutions for combating vector-borne diseases, managing crop pests, and supporting ecosystem conservation efforts. Current technologies raise safety concerns for unintended gene propagation. Herein, we address such concerns by splitting the drive components, Cas9 and gRNAs, into separate alleles to form a trans-complementing split–gene-drive (tGD) and demonstrate its ability to promote super-Mendelian inheritance of the separate transgenes. This dual-component configuration allows for combinatorial transgene optimization and increases safety by restricting escape concerns to experimentation windows. We employ the tGD and a small–molecule-controlled version to investigate the biology of component inheritance and resistant allele formation, and to study the effects of maternal inheritance and impaired homology on efficiency. Lastly, mathematical modeling of tGD spread within populations reveals potential advantages for improving current gene-drive technologies for field population modification.

[1] Section of Cell and Developmental Biology, University of California San Diego, La Jolla, CA 92093, USA. [2] Division of Epidemiology and Biostatistics, School of Public Health, University of California, Berkeley, CA 94720, USA. [3] Biophysics Graduate Group, University of California, Berkeley, CA 94720, USA. [4] Tata Institute for Genetics and Society, University of California, San Diego, 9500 Gilman Drive, La Jolla, CA 92093-0349, USA. *email: vgantz@ucsd.edu

CRISPR gene-drive systems offer tremendous potential for engineering wild populations due to their ability to self-propagate, biasing inheritance from Mendelian (50%) to super-Mendelian (>50%)[1–7]. This technology has important applications in fighting vector-borne diseases (e.g., malaria) by suppressing[3,7] or modifying[4] mosquito populations to decrease their burden on public health, managing crop pests[8,9], and suppressing invasive rodents to support island restoration efforts[10,11]. While the scientific community welcomes the enormous promise of this technology for solving significant global issues, it also acknowledges that it is currently in its infancy and that several gaps need to be filled before it can be safely deployed[12–14]. In particular, concerns have been raised about accidental release during laboratory research or premature release of insects into the field, highlighting a need for the development of strategies to increase safety during optimization phases[15].

Gene-drive systems use an allelic conversion process that occurs in the germline, changing heterozygous to homozygous cells that can achieve the super-Mendelian inheritance necessary for population engineering. Currently, two different approaches based on the RNA-guided endonuclease Cas9 have been experimentally evaluated: (1) a full gene drive (full GD) is the traditional format, consisting of a Cas9 and a guide RNA (gRNA) gene inserted at the target location as a single unit. The two gene products combine to induce a double-strand break at the same position on the wild-type allele, which is then repaired via homology-directed repair (HDR) using the intact chromosome carrying the gene-drive element as a template[2–4,7,16–18]. (2) The gRNA-only gene drive (gRNA GD) is based on CopyCat gRNA elements[19] that are capable of allelic conversion in the presence of a separate genetic source of Cas9. Since only the gRNA element is propagated in this case, its spread is regulated by the presence of a separate, static Cas9 transgene[10,19–21]. The use of a full GD is causing concern to the scientific community as an accidental release could spread unchecked[15]. While a gRNA GD would address such concerns, its application in the field for large-scale population engineering is unlikely to succeed since it would require that a large percentage of the population carried a Cas9 transgene[22].

Here, we develop a CRISPR gene-drive method in *Drosophila* called trans-complementing gene drive (tGD), which combines the strengths of both approaches described above. This arrangement splits the Cas9 and gRNA into two different transgenic lines. When separated, neither component displays gene-drive activity, providing the same safety profile of a gRNA-only drive. When combined by genetic cross, however, the two complementary components reconstitute the properties of a full GD, resulting in both elements propagating together. Here, we demonstrate that a tGD system can bias the inheritance of two interdependent transgenes, and that such a split arrangement can be used to deconstruct specific drive parameters. We exploit the modularity of the tGD to dissect specific features influencing gene-drive efficiency: (1) the functionality of different Cas9 promoters and their maternal effect on the tGD that has been shown in other systems to be a potential source of resistance[2,4,16,17], (2) how genomic context can affect gene-drive efficiency, and (3) the effect of impairing homology between the drive construct and the targeted allele. In addition, we apply a drug-regulation technology to the tGD system such that super-Mendelian inheritance can be controlled by the presence of a small molecule in the fly diet, and use this tool to restrict Cas9 activation in the adult germline and study gene-drive function in this tissue. Last, we simulate the propagation of tGD elements and uncover their potential to spread transgenes to a higher fraction of a population than a corresponding full-drive system, highlighting the tGD's potential for future field applications.

## Results

**The tGD system displays super-Mendelian behavior.** The tGD system was designed to split the two genetic elements, Cas9 and a two-part gRNA gene construct (gRNA-A and gRNA-B), into two distinct genomic locations, which when separated, behave as regular Mendelian transgenes (no gene-drive activity) (Fig. 1a). Once combined by genetic crossing, gRNA-A cleaves the genome at the Cas9 integration site while gRNA-B cuts at the gRNA locus (Fig. 1a). Because the cleaved ends match with perfect homology to the sequences flanking each of the transgenic elements, the HDR pathway inserts a copy of each transgene into the wild-type allele (Fig. 1a).

As a first test of this system, we generated the tGD(*y*,*e*) targeting the coding sequences of the *yellow* (*y*) and *ebony* (*e*) loci. Loss-of-function mutations in either of these genes result in whole-body pigmentation phenotypes, lighter and darker, respectively, and are therefore readily detectable[23]. To accomplish this goal, we introduced a *Streptococcus pyogenes* *Sp*Cas9 (Cas9) source driven by the *vasa* promoter into the *yellow* gene (X chromosome) marked with a DsRed (Red) fluorescent reporter expressed in the eye to generate the *vasa*-Cas9 line (Supplementary Fig. 1). We placed the second transgene, carrying the gRNA tandem cassette (gRNA-*y1* and gRNA-*e1*), on chromosome III (autosome), disrupting the *ebony* gene. This cassette was instead marked with EGFP (Green) to generate the *e*-[*y1*,*e1*] line (Supplementary Fig. 1).

We tested the tGD(*y*,*e*) arrangement by individually crossing *vasa*-Cas9 males to *e*-[*y1*,*e1*] virgin females (F$_0$, Fig. 1b) and collecting F$_1$ transheterozygous virgin females carrying both constructs. These females were single-pair mated to wild-type Oregon-R (Or-R) males (F$_1$, Fig. 1b). Phenotypic analysis of the fluorescent markers in the resulting F$_2$ progeny allowed simultaneous evaluation of the germline output inheritance rates of both the Cas9 and gRNA transgenes of each single F$_1$ female (F$_2$, Fig. 1b). We scored the F$_2$ progeny of 11 F$_1$ females and observed an inheritance of greater than 50% for both transgenes, with an average of 83% and 85% for Cas9-Red and gRNA-Green, respectively (Fig. 1c, Supplementary Data 1). In addition, we tested the tGD(*y*,*e*) arrangement using a *nanos*-Cas9 construct inserted at the same *yellow* locus (Supplementary Fig. 1) and observed similar inheritance rates of the two elements (Supplementary Data 1). Since the gRNA-Green transgene is targeting the *ebony* gene, which is located on an autosome (Chromosome III), allelic conversion could occur both in females and males at this location. To test how the gRNA-Green transgene would perform in the male germline, we crossed F$_1$ males (instead of F$_1$ females) carrying both Cas9-Red and gRNA-Green transgenes to our wild-type strain females to score the inheritance rate in the F$_2$ progeny. In this experiment, the gRNA-Green transgene displayed a 67% average inheritance. Interestingly, allelic conversion in F$_1$ males was lower than that observed through the female germline (67% vs. 85%, respectively; see statistical analysis in Supplementary Data 1). Similar observations were made in a previous study using a full gene drive targeting the *cinnabar* gene, also located on an autosome[17]. The Cas9-Red transgene carried on the X chromosome of F$_1$ males was, as expected, inherited in a Mendelian fashion (~50%). Males have only a single X chromosome; therefore, no gene conversion events are possible in this situation at the *yellow* locus. Indeed, the Cas9-Red construct was inherited by all F$_2$ daughters, while all males inheriting the Y chromosome were DsRed negative (Supplementary Data 1).

The above results demonstrate that a CRISPR gene drive can be split into two separate genetic elements located on different chromosomes, which once combined, can be simultaneously propagated with super-Mendelian inheritance. This conditional

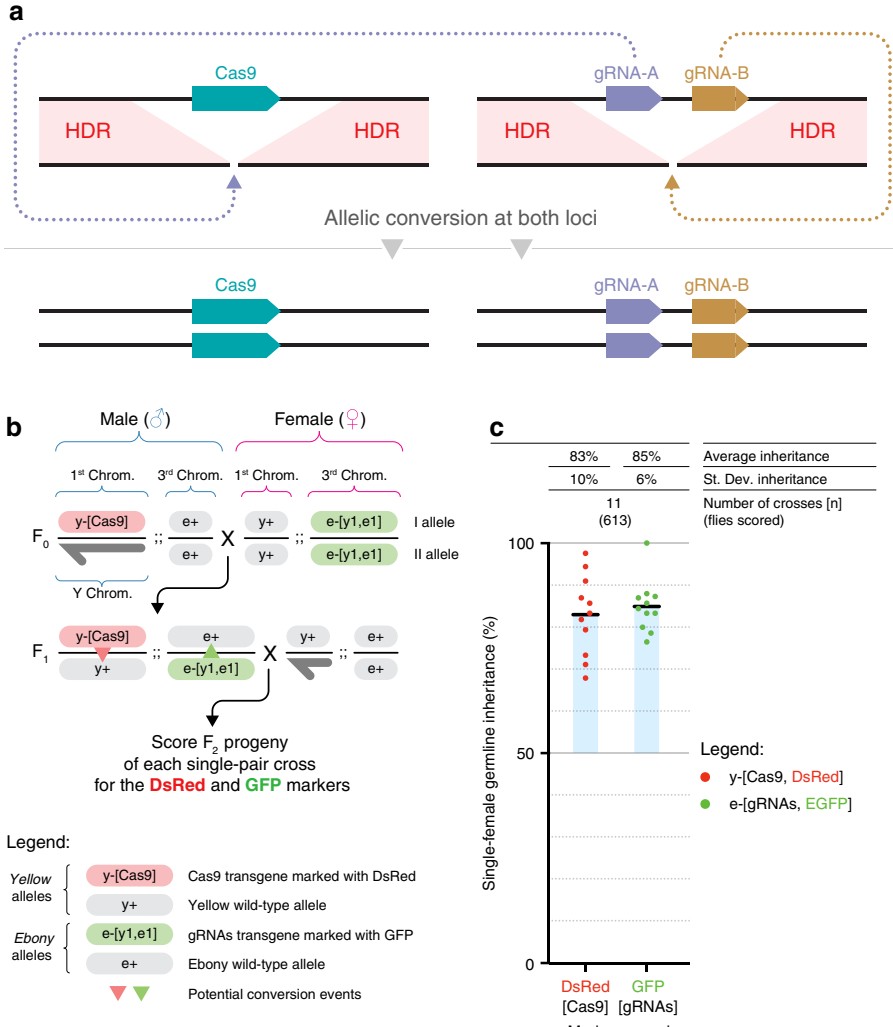

**Fig. 1 The tGD system allows simultaneous super-Mendelian inheritance of two transgenes. a** Schematic of the tGD genetic arrangement with two elements that can be kept separated as different transgenic lines. The Cas9 transgene is inserted in the genomic location targeted by the gRNA-A, while a separate cassette expressing a tandem-gRNA construct (gRNA-A, gRNA-B) is inserted at the location targeted by gRNA-B. Upon genetic cross, each of the gRNA-A and gRNA-B combines with Cas9 to generate a double-strand DNA break at each locus on the wild-type allele (dotted purple and brown arrows, respectively). Each break is then repaired by the homology-directed repair (HDR) pathway using the intact chromosome carrying the transgene as a template. **b** Outline of the genetic cross used to demonstrate tGD in fruit flies, indicating transgenes and wild-type allele location on different chromosomes. Sex of the individuals is indicated with symbols "♂" for males and "♀" for females. $F_0$ males carrying a DsRed-marked Cas9 transgene inserted into the *yellow* locus were crossed to females carrying a GFP-marked cassette containing two gRNAs (*y1–e1*) inserted into the *ebony* coding sequence. Transheterozygous $F_1$ females (carrying both Cas9 and gRNAs) were crossed to wild-type males to assess germline transmission rates of the fluorophores marking the transgenes in the $F_2$ progeny. The conversion events are indicated by the red and green triangles in the $F_1$ females. **c** Single $F_1$ female germline inheritance output is measured as GFP and DsRed marker presence in the $F_2$ progeny. The black bar represents the inheritance average. The blue shading represents the deviation from the expected 50% "Mendelian" inheritance. Inheritance average, standard deviation, number of samples (*n*), and total number of flies scored in each experiment are represented over the graph in line with the respective data. Raw phenotypical scoring is provided as "Supplementary Data 1".

property offers flexibility and increases safety while functioning as a full GD. Furthermore, these findings have implications for other strategies that similarly use multiple elements driving simultaneously such as the proposed "daisy-chain drive"[24] or integral gene drives[25].

We next confirmed the tGD strategy in which the gRNA component was inserted at different chromosomal sites, namely the *white* (*w*) locus[1,2,17]. We tested the tGD(*y,w*) by constructing an EGFP-tagged tandem-gRNA element (*w-[y1,w2]*) targeting both *y* and *w* (Supplementary Fig. 1). *w-[y1,w2]* virgin females ($F_0$) were crossed to *y*-inserted *vasa*-Cas9 males, and $F_1$ virgin females were collected and outcrossed to wild-type Or-R males (Fig. 2a). In the

$F_2$ progeny, we observed 89 and 96% inheritance rates of the Cas9-Red and gRNA-Green transgenes, respectively (Fig. 2e; Supplementary Data 2), which were both higher transmission rates than when the gRNA construct was inserted into the *ebony* locus. Interestingly, the same Cas9-Red transgene displayed a significantly higher inheritance rate in the tGD(*y,w*) (89%) than in the tGD(*y,e*) (83%) (statistical analysis in Supplementary Data 2). This difference might result from positional effects modulating the *y1*-gRNA expression when inserted at a different genomic location (*white* or *ebony*, respectively), or perhaps reflects the distance of the transgenes in the two systems: tGD(*y,w*) are close together on the same chromosome, while tGD(*y,e*) are on different

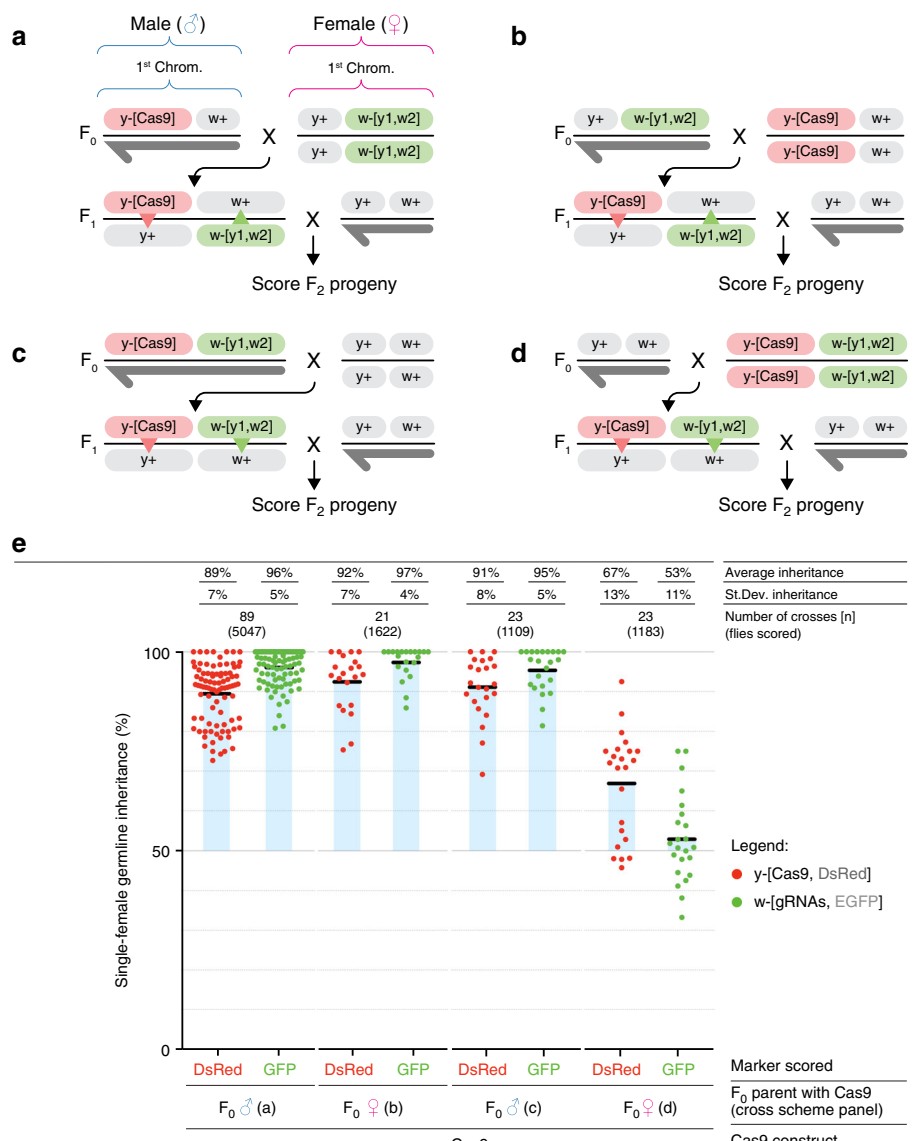

**Fig. 2 tGD targeting the *yellow* and *white* loci uncovers the maternal effect mechanism. a–d** Genetic crosses performed using the tGD elements targeting *yellow* (DsRed-Cas9) and *white* (GFP-*y1,w2*-gRNAs) loci, both located on the first chromosome (X chromosome). Cross schemes used to test the $F_1$ female germline by scoring the $F_2$ progeny: **a** Cas9 from the $F_0$ male and gRNAs from the $F_0$ female, **b** Cas9 from the $F_0$ female and gRNAs from the $F_0$ male, **c** both Cas9 and gRNAs from the $F_0$ male, and **d** both Cas9 and gRNAs from the $F_0$ female. Allelic conversion events are indicated by the red and green triangles in the $F_1$ females of every cross scheme. **e** Analysis of the $F_2$ inheritance rates of the fluorescent markers for all cross scheme combinations (**a–d**) using Cas9 constructs driven by the *vasa* promoter. Strong super-Mendelian inheritance is seen for all conditions except when both Cas9 and the gRNAs are inherited from the $F_0$ female (**d**). Values for the inheritance average (black bar), standard deviation, number of samples (*n*), and total number of flies scored in each experiment are represented over the graph in line with the respective data. Sex of the $F_0$ parent carrying Cas9 is indicated with "♂" for males and "♀" for females in panel (**e**). Raw phenotypic scoring is provided as "Supplementary Data 2".

chromosomes. We further tested the impact of genomic location on allelic conversion efficiency by swapping the locations of the tGD(*y,w*) transgenes, placing the Cas9 element in *white* and the gRNAs in *yellow* to generate tGD(*w,y*) (Supplementary Fig. 1). This swapped configuration displayed 95 and 98% conversion efficiency for the Cas9 and gRNA constructs, which are significantly higher rates than those observed for tGD(*y,w*) (89% at *yellow* and 96% at *white*; Supplementary Fig. 2, statistical analysis in Supplementary Data 2). The combined tGD(*w,y*) systems also copy with higher conversion rates than those observed in previous studies using full GDs at the *y* and *w* loci[2,17,21]. These greater copying efficiencies may reflect differing gRNA efficiencies[26,27], or genetic background effects associated with our Or-R strains vs. the *w[1118]* and Canton-S strains used in previous studies[2]. Regardless, these results

suggest that conversion efficiencies are impacted by the genomic location of the transgene and that transgene size in the range tested (3–8.3 kbp) does not seem to negatively affect allelic conversion in our system.

**X-chromosome tGD uncovers the maternal effect on inheritance.** Previous studies reported that the inheritance of a gene drive from the female germline could lead to the generation of early embryogenesis mutations[2,4,17], which occur when cleaved-allele repair results in small insertion/deletions (indels) at the cut site instead of allelic conversion, through alternative repair pathways such as nonhomologous end joining (NHEJ)[28]. Indel alleles therefore represent an obstacle for CRISPR gene-drive propagation in subsequent generations; as such alleles would be

resistant to the gene-drive action, preventing its spread[2,4,16,17]. These resistant alleles are generated at a high rate when a gene drive is inherited from a female parent, most likely due to Cas9 deposition in the egg[2–4,17]. We therefore performed the reciprocal cross using our tGD($y,w$) approach by collecting $F_0$ vasa-Cas9 females and gRNA-carrying males (Fig. 2b). In the $F_2$ analysis, we observed similar inheritance rates to the previous tGD analysis of 92% (Cas9-Red) and 97% (gRNA-Green) (Fig. 2e, Supplementary Data 2), indicating no reduction in inheritance rates when Cas9 but not the gRNA is received from the mother in the tGD arrangement, paralleling observations from a gRNA-only drive system[21], and differing from what was reported in previous full-GD approaches[2–4,17].

Since the two CRISPR components of our tGD are inherited separately, we used this system to test whether both components had to be simultaneously deposited in the egg to observe the maternal effect. We generated a homozygous line carrying both elements on the same chromosome to analyze the allelic conversion efficiency for co-inheritance. As before, we analyzed the $F_2$ progeny of cross schemes in which the coupled Cas9/gRNA elements were inherited together from either the $F_0$ male (Fig. 2c) or female (Fig. 2d) in a configuration mimicking a full-GD scenario. Here, the $F_2$ progeny from $F_0$ male inheritance had inheritance rates of 91% (Cas9) and 95% (gRNA) (Fig. 2e). In contrast, the inheritance rates from the $F_0$ female were 67% (Cas9) and 53% (gRNAs) (Fig. 2e), suggesting that a strong maternal effect on a gene drive is generated only when the two elements are inherited together from a female germline, consistent with previous observations[21]. These results encourage new gene-drive designs that delay Cas9 and/or gRNA functioning in the embryo to avoid the undesired generation of drive-resistant alleles.

We next tested how the tGD($y,w$) would perform in terms of inheritance rates and maternal effects when using a different promoter to drive expression of Cas9. We cloned the nanos gene regulatory region into our Cas9 construct and inserted it into the same genomic location (yellow) to generate the nanos-Cas9 line (Supplementary Fig. 1). Performing the same cross schemes to combine the nanos-Cas9-Red with the $y$-[y1,w2] line (Fig. 2a–d), a similar pattern was seen for the vasa promoter, with comparable inheritance rates for separated transgenes in the $F_0$ crosses and frequent formation of resistant alleles when the combined Cas9/gRNA complex was inherited from $F_0$ females (Supplementary Fig. 3, Supplementary Data 2). An additional noteworthy result from this experiment was that our tGD($y,w$) driven by nanos did not display the inheritance differences between the Cas9 and gRNAs as noted in our previous vasa-driven tGD($y,w$) experiment in which both elements were inherited separately (Fig. 2a, e; Supplementary Fig. 3; statistical analysis in Supplementary Data 2). These results reinforce the hypothesis that inheriting a preloaded Cas9–gRNA complex through the mother is an obstacle to the spread of gene drives, which did not occur when the elements were inherited separately in our system.

**tGD generates predictable resistant alleles**. To better understand the maternal effect on gene-drive inheritance, we used males recovered from the $F_2$ generation of the simultaneous-inheritance tGD($y,w$) crosses that carried resistant alleles. Since males have only a single X chromosome, which is inherited from the $F_1$ female, they are suitable for phenotypic isolation and molecular characterization of non-conversion events (resistant alleles) that occurred in each single $F_1$ female. We sequenced resistant alleles in $F_2$ males from various experimental conditions selecting, from 57 (white) and 60 (yellow) independent $F_1$ female germlines, a

total of 242 and 225 flies per locus, respectively (Fig. 3a, b, Supplementary Fig. 4). Intriguingly, for both $w$ and $y$, we recovered three resistant alleles that occurred repeatedly in the germline of independent $F_1$ females and that covered the majority of all sequenced flies (named $w$A, $w$B, and $w$C and $y$A, $y$B, and $y$C; Fig. 3a, b, Supplementary Fig. 4). In addition, we observed fewer unique occurrences of other indels in white than in yellow, 18/242 (7%) and 43/225 (19%), respectively (Fig. 3a, b, light gray). These findings highlight the importance of characterizing the range of possible indels at genomic locations chosen for field gene-drive applications. In addition, we analyzed the frequency of resistant mutations generated from conditions meant to resemble full-GD situations, with both elements inherited simultaneously (Fig. 2c, d), and observed that the ratios and the type of recurrent alleles recovered can vary drastically between different drive configurations (Supplementary Fig. 5).

Lastly, in an attempt to clarify at what time point during development the allelic conversion process occurs, we also tabulated the number of different resistant alleles recovered in the $F_2$ progeny of single $F_1$ females. Under the analyzed conditions, we detected a range of 1–4 different resistant alleles per vial (Fig. 3c–j). When the Cas9 and gRNA constructs were inherited from the $F_0$ female (Fig. 2d), we recovered only one resistant white allele per vial analyzed for both vasa and nanos promoters (Fig. 3d, f), in line with previous observations using a full gene drive driven by nanos[17]. Our results suggest that these indels are generated as early as fertilization (or zygote), consistent with the average inheritance of ~50% observed (vasa Fig. 2d, e and nanos Supplementary Fig. 3). This trend was not observed for the yellow locus, in which up to three different alleles were recovered under the same conditions, suggesting that the promoter used to express the gRNAs or the gRNA itself results in lower efficiency of cutting at the yellow locus than for white (Fig. 3h, j). While previous work compiled multiple NHEJ sequences by directly sequencing $F_1$ females[1,2], our results expand on these observations by showing different resistant alleles are generated within the germline by tracking the individual genotypes of multiple $F_2$ progeny.

Regarding the male inheritance (Fig. 2c), we observe 1–4 indels generated from each $F_1$ female germline for either promoter and locus analyzed (Fig. 3c, e, g, i). This fact, combined with the high inheritance rates observed in these experiments, suggests a model in which resistant alleles are stochastically generated during late germline development, following pole-cell formation (Fig. 4a). In addition, under these conditions, adult $F_1$ females display $w$-mosaic eyes due to leakiness of the vasa and nanos promoters in somatic tissues, suggesting that the somatic tissue was not previously edited during early embryogenesis (Fig. 4a). These observations differ from previous work using nanos where no mosaicism was detected when the drive was inherited through the male[17]. These discrepancies could derive from the use of different codon usage for the Cas9 sequences.

Conversely, in female inheritance conditions, the indels seem to be generated in the syncytial blastoderm embryo before cellularization of the ~3 primordial pole-cell precursors that have been estimated to be set aside for the germline at the 128-cell stage[29]. In fact, at the white locus, we observed only a single indel per female germline, which combined with the ~50% inheritance measured for the gRNA transgene and the fully $w$- $F_1$ female phenotype, suggests that Cas9 action occurs efficiently at the zygote stage (Fig. 4b). For yellow, we instead observed 1–3 indels generated per $F_1$ female, suggesting that Cas9 cutting happens later or less efficiently prior to pole-cell formation. In the case of yellow, our data are also consistent with a few nuclei escaping Cas9 action in the blastoderm embryo and being converted later during germline development as we observe super-Mendelian

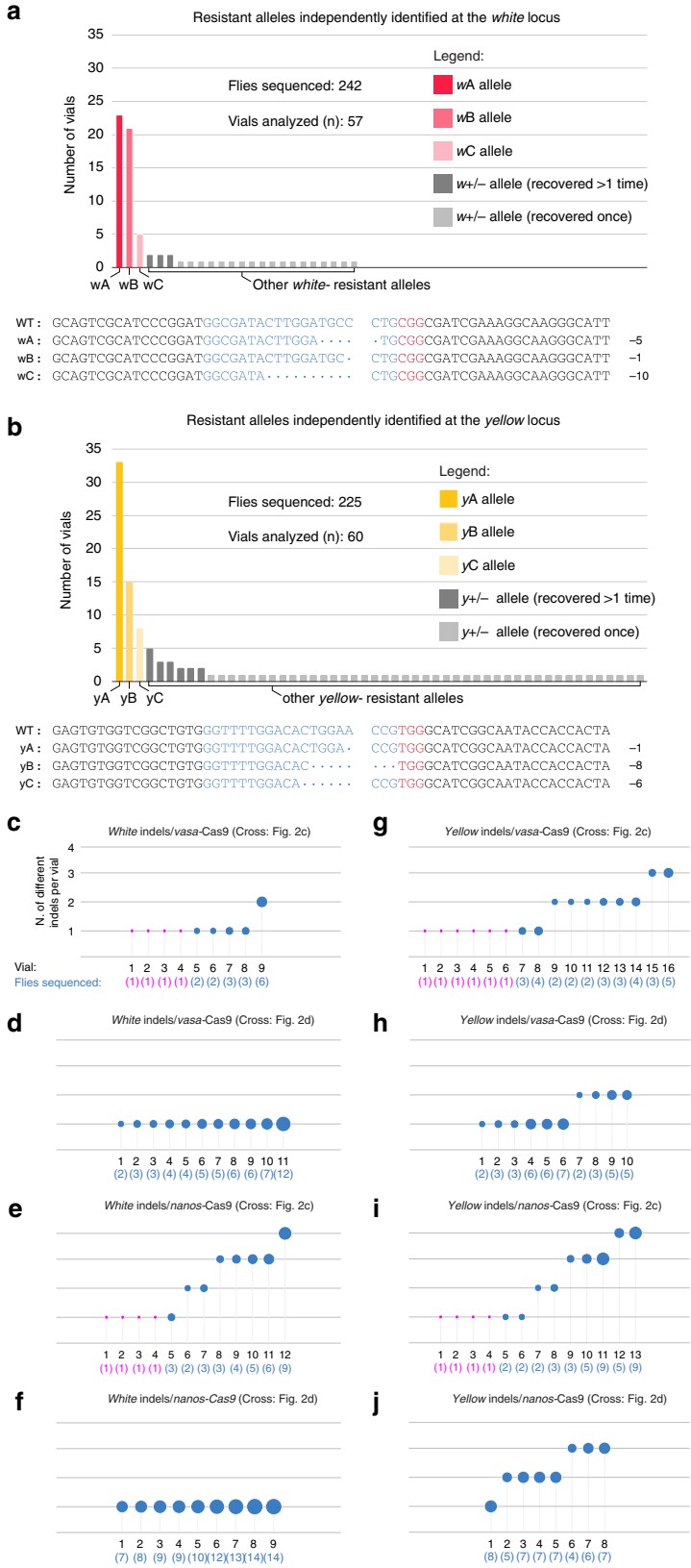

**a** Resistant alleles independently identified at the *white* locus

Flies sequenced: 242

Vials analyzed (n): 57

Legend:
- *w*A allele
- *w*B allele
- *w*C allele
- *w*+/− allele (recovered >1 time)
- *w*+/− allele (recovered once)

wA wB wC    Other *white*- resistant alleles

```
WT : GCAGTCGCATCCCGGATGGCGATACTTGGATGCC  CTGCGGCGATCGAAAGGCAAGGGCATT
wA : GCAGTCGCATCCCGGATGGCGATACTTGGA····  ·TGCGGCGATCGAAAGGCAAGGGCATT  −5
wB : GCAGTCGCATCCCGGATGGCGATACTTGGATGC·  CTGCGGCGATCGAAAGGCAAGGGCATT  −1
wC : GCAGTCGCATCCCGGATGGCGATA··········  CTGCGGCGATCGAAAGGCAAGGGCATT  −10
```

**b** Resistant alleles independently identified at the *yellow* locus

Flies sequenced: 225

Vials analyzed (n): 60

Legend:
- *y*A allele
- *y*B allele
- *y*C allele
- *y*+/− allele (recovered >1 time)
- *y*+/− allele (recovered once)

yA yB yC    other *yellow*- resistant alleles

```
WT : GAGTGTGGTCGGCTGTGGGTTTTGGACACTGGAA  CCGTGGGCATCGGCAATACCACCACTA
yA : GAGTGTGGTCGGCTGTGGGTTTTGGACACTGGA·  CCGTGGGCATCGGCAATACCACCACTA  −1
yB : GAGTGTGGTCGGCTGTGGGTTTTGGACAC·····  ···TGGGCATCGGCAATACCACCACTA  −8
yC : GAGTGTGGTCGGCTGTGGGTTTTGGACA······  CCGTGGGCATCGGCAATACCACCACTA  −6
```

**c** *White* indels/*vasa*-Cas9 (Cross: Fig. 2c)

N. of different indels per vial

Vial: 1 2 3 4 5 6 7 8 9
Flies sequenced: (1)(1)(1)(1)(2)(2)(3)(3)(6)

**g** *Yellow* indels/*vasa*-Cas9 (Cross: Fig. 2c)

1 2 3 4 5 6 7 8 9 10 11 12 13 14 15 16
(1)(1)(1)(1)(1)(3)(4)(2)(2)(3)(3)(4)(3)(5)

**d** *White* indels/*vasa*-Cas9 (Cross: Fig. 2d)

1 2 3 4 5 6 7 8 9 10 11
(2)(3)(3)(4)(4)(5)(5)(6)(6)(7)(12)

**h** *Yellow* indels/*vasa*-Cas9 (Cross: Fig. 2d)

1 2 3 4 5 6 7 8 9 10
(2)(3)(3)(6)(6)(7)(2)(3)(5)(5)

**e** *White* indels/*nanos*-Cas9 (Cross: Fig. 2c)

1 2 3 4 5 6 7 8 9 10 11 12
(1)(1)(1)(1)(3)(2)(3)(3)(4)(5)(6)(9)

**i** *Yellow* indels/*nanos*-Cas9 (Cross: Fig. 2c)

1 2 3 4 5 6 7 8 9 10 11 12 13
(1)(1)(1)(1)(2)(2)(2)(3)(5)(5)(9)(5)(9)

**f** *White* indels/*nanos*-Cas9 (Cross: Fig. 2d)

1 2 3 4 5 6 7 8 9
(7)(8)(9)(9)(10)(12)(13)(14)(14)

**j** *Yellow* indels/*nanos*-Cas9 (Cross: Fig. 2d)

1 2 3 4 5 6 7 8
(8)(5)(7)(7)(7)(4)(6)(7)

inheritance in the offspring of some F$_1$ females (Figs. 4c, 2d, e; Supplementary Fig. 3).

While our findings are not conclusive, all cases of female inheritance analyzed here and reported in previous studies using comparable reagents[1,4,16,17] support the hypothesis that

Cas9-mediated cleavage leads either to copying via gene conversion or to the generation of resistant alleles. The results also suggest that such events occur during early embryogenesis, perhaps at differing stages, but prior to formation of the mature germline. Importantly, our system allowed us to evaluate, in the

**Fig. 3 Analysis of resistant allele formation in the tGD(*y*,*w*) configuration. a**, **b** Graphs represent the independent generation of specific indel mutations generated in the experiments carried out in Fig. 2 and Fig. S3 when allelic conversion failed at the (**a**) *white* and (**b**) *yellow* loci. At both loci, we observe three repeatedly isolated indels that are colored with different shades of red for the *white* locus (*w*A, *w*B, and *w*C, dark, medium, and light red, respectively) and yellow for the *yellow* locus (*y*A, *y*B, and *y*C, dark, medium, and light yellow, respectively), the sequence of which is reported under each graph indicating with dots the missing bases compared with the wild-type sequence, split at the expected cut site. Additional indels recovered more than once are colored in dark gray, and those recovered only once are colored in light gray. **a**, **b** Legends describing symbolism are overlaid on each of the two graphs. **c–j** Each panel depicts the number of different indels recovered (*y*-axis) in ascending order for each $F_1$ female (numbered on *x*-axis) for which the $F_2$ male progeny was sampled. Blue bubble size indicates the number of flies analyzed for each specific $F_1$ female, which is also reported under the vial number in parenthesis. For $F_1$ females producing only one $F_2$ male with an indel, and therefore only one male was sampled, the bubble is represented in magenta instead of blue.

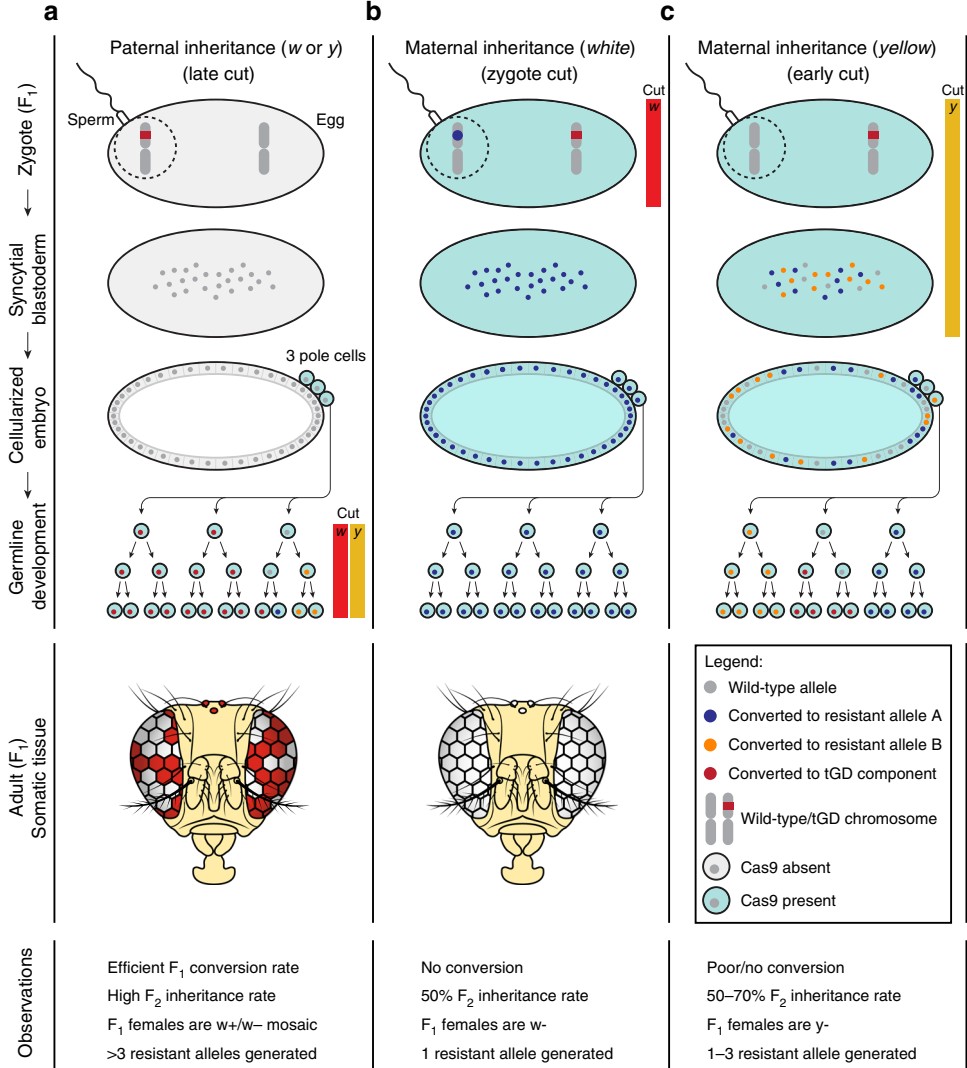

**Fig. 4 Model of tGD transgene behavior in males and females. a–c** Schematic representation of the different scenarios observed in our tGD experiments when both elements (Cas9 and gRNA transgenes) are inherited together. **a** When the tGD chromosome (labeled in red) comes from the dad (sperm) any Cas9 activity was detected from zygote to cellularized embryo. The high $F_2$ inheritance rates in these conditions suggest that Cas9 action (red and yellow bars) is restricted to the germline development phase for the *white* and *yellow* genes, allowing efficient conversion (red dots) in this stage. Since the conversion process would occur after three pole-cell formation, this would be consistent with the 1–4 resistant allele range (blue and orange dots) captured from $F_1$ germline females analyzed. **b**, **c** We detected different outputs for both *yellow* and *white* loci when the tGD chromosome was inherited from the mother. **b** Cas9 cut (red bar) seems to occur at the zygote stage for the *white* locus since we only observe one resistant allele per each $F_1$ germline female that was analyzed. Consistently, this early cleavage event leads to 50% inheritance of our transgene precluding any further conversion event. **c** Cleavage of the *yellow* locus by Cas9 (yellow bar) seems to be delayed based on the super-Mendelian inheritance observed in the scored $F_2$ flies from some $F_1$ germline females. In this case, the Cas9 activity and the conversion process could occur in the zygote or syncytial blastoderm stage, and before the pole-cell establishment. This is reinforced by the the observed range of 1–3 resistant alleles (blue and orange dots).

same animal, the simultaneous action of two gRNAs and identify how they differ in the maternal effect on super-Mendelian inheritance (Fig. 4).

**Controlled tGD activation in the adult germline**. Our above-presented studies on female germline resistance suggest that the Cas9-induced cleavage events could happen as early as the zygote stage. As resistant alleles pose a potential problem to gene-drive applications, we wondered how tGD would perform when Cas9 activity was solely restricted to the adult germline. This is an important question since, to our knowledge, no published gene-drive work has thus far been able to precisely establish the timing of drive conversion events.

We recently developed a small-molecule-controlled system for use in active genetics approaches including CRISPR-based gene-drive systems[30]. Briefly, we fused *Escherichia coli* dihydrofolate reductase domains to *Sp*Cas9 to promote its rapid proteasomal degradation in the absence of the stabilizing small molecule trimethoprim (TMP)[31,32]. We showed that TMP addition to the fruit fly diet stabilized our modified *Sp*Cas9 (DD2-Cas9) and sustained super-Mendelian inheritance control of a CopyCat active genetic element[30].

Here, we first used a comparable DD2-Cas9 line and showed that the mentioned drug-regulated system could be applied to the tGD(*y*,*w*) for controlling its super-Mendelian inheritance (Supplementary Fig. 6; Supplementary Data 3). Next, we used the TMP regulation in our tGD system and were able to activate Cas9 only in the adult female germline, showing that super-Mendelian inheritance can be achieved when the gene-drive process is restricted to this tissue (Supplementary Fig. 6; Supplementary Data 3), although resistant alleles were also detected (Supplementary Fig. 7).

This approach opens a new avenue for restricting Cas9 activity to an optimal window when HDR is favored, perhaps representing a way to bypass the maternal effect. In addition, future developments of this technology could bias inheritance, for example, in a spatially restricted fashion, such as by city, through the addition of the small molecule to urban water reservoirs, therefore controlling the spread of a gene drive into a circumscribed locale.

**Impaired homology asymmetrically affects drive efficiency**. Recent research has raised concerns that natural polymorphisms could also hamper gene-drive spread in heterogeneous populations[33]. A proposed strategy to increase drive efficiency and work around resistant alleles or polymorphisms is to use multiple gRNAs to ensure cutting and lower the chances of an indel[17,34,35]. However, in such scenarios one cannot achieve perfect homology with all possible DNA ends generated when using multiple gRNAs. For example, when using two gRNAs, one cut can be generated earlier and repaired by NHEJ generating an indel at that location. Subsequent cutting by the second gRNA would generate a repair template carrying a nonhomologous overhang on one side and perfect homology on the other. We wondered to what extent such potential homology discordance between the cleaved chromosome and the allele to be propagated would affect the efficiency of a gene drive. We reasoned that our tGD system would be an ideal tool to test this hypothesis, by impairing homology on the gRNA construct while leaving untouched the *vasa*-Cas9 element as an internal control.

For this purpose, we generated three modified versions of our *w*-[*y1*,*w2*] line that varied the location of a 20 bp lack in sequence homology: i) the first lacked 20 bp of homology on each side to generate the *w*-[*y1*,*w2*]-Truncated-HA line (both sides impaired), ii) the second lacked 20 bp on the side of the

protospacer-adjacent motif (PAM), which is an essential DNA-homing sequence for CRISPR function, side of the gRNA (PAM proximal), and iii) the third lacked 20 bp on the side distal to the PAM (PAM distal) (Fig. 5, Supplementary Fig. 1). We performed the crosses according to the same scheme as in Fig. 2a by combining the *vasa*-Cas9 line with each of the three lines bearing impaired homology and scoring the F₂ progeny for inheritance rates.

When both homology arms were impaired, we observed a significantly lower gRNA transgene inheritance, average of 72% (Fig. 5a, first condition; statistical analysis in Supplementary Data 4), than the previously observed 96% with perfect homology (Fig. 2e, first condition). Notably, when we examined drive in the unilaterally deleted PAM-proximal and PAM-distal lines, we only observed a significant decrease when the PAM-distal homology was impaired (Fig. 5a; statistical analysis in Supplementary Data 4). Our internal control Cas9 transgene averaged ~90% inheritance rates in all conditions. Interestingly, when homology on both sides was impaired, we noted a slightly lower rate of inheritance for the Cas9 transgene. While the difference in average inheritance was not significant, the distribution of data points appeared to be altered as reflected by an increased standard deviation (Fig. 5a; statistical analysis in Supplementary Data 4). These results indicate that gene-drive applications with inherent imperfect homology are feasible, although the nature of the homology should be considered when designing multiplexed gRNA strategies. To further explore the impact of the homology, we compared plots showing the correlation between the inheritance rates for allelic conversion at the two loci, Cas9-red on the Y axis and gRNA-Green on the *x*-axis (Fig. 5c–f); we observe that when the homology is lacking on the PAM-proximal side, the data distribution resembles the pattern for the perfect homology construct, with the majority of dots located just over the diagonal (Fig. 5c). In contrast, an equivalent distribution of dots is observed on both sides of the diagonal when the homology is lacking on the PAM-distal side (Fig. 5f). Indeed, when comparing the fraction of dots below the diagonal, we observed a significant difference when the PAM-distal side was impaired (statistical analysis in Supplementary Data 4). It has been shown that Cas9 can remain bound to the DNA after cleavage for an extended time[36–38] and preferentially releases the PAM-distal, nontarget strand[39]. Our data suggest that this skewed outcome might result in an asymmetrical influence of the bound Cas9/gRNA complex on the HDR process, and that the released PAM-distal, nontarget strand may promote efficient HDR. This property might be harnessed for increasing HDR in systems that currently display poor efficiency, such as human somatic cells in therapeutic efforts[40]. Regarding the use of multiplexing for gene drives, our data suggest that it should be most efficient to use only two gRNAs with PAMs pointing toward each other (PAM-in), at least when generating cuts within the tested range (20 bp). Increasing the distance between the two cleavage sites would generate longer overhangs and could result in altered efficiency. Using three or more gRNAs would, in certain situations, generate repair templates with two nonhomologous overhangs that should dramatically lower HDR-mediated repair.

**Modeling of tGD predicts benefits for field deployment**. We performed mathematical modeling to predict the extent of tGD spread compared with full GDs and to determine tGD suitability for population replacement in the field. Four tGD systems were considered: (i) components linked on an autosome (tGD), (ii) components unlinked at two autosomal loci (tGDc), (iii) components linked on the X chromosome (tGDX), and (iv) components unlinked on the X chromosome (tGDXc). We also

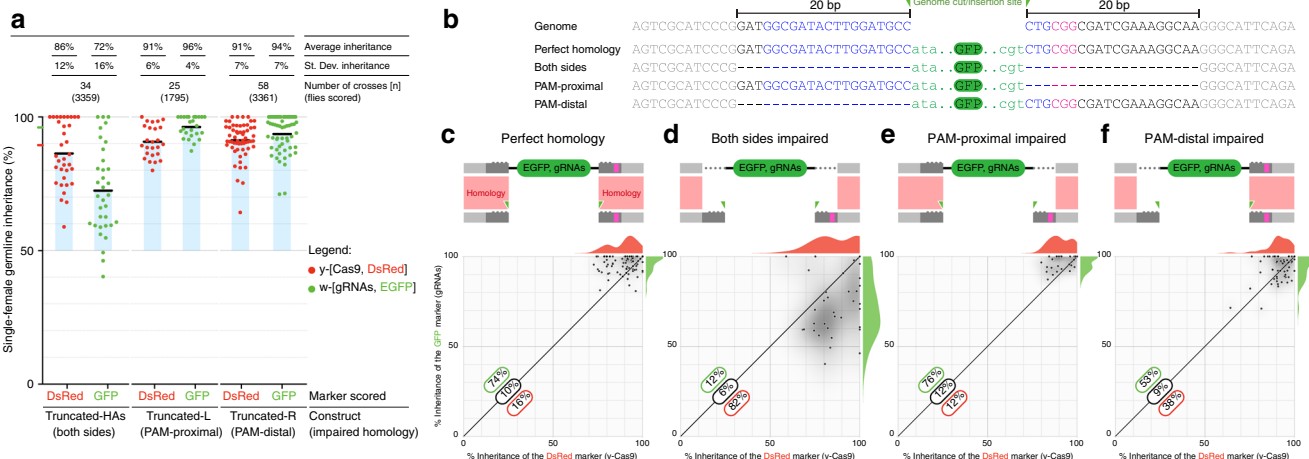

**Fig. 5 Impaired homology arms affect drive efficiency. a** Inheritance rates of the three modified w-[y1,w2] gRNA lines (combined with vasa-Cas9 in yellow) in which the homology (**b**) was impaired by removing 20 bp of sequence on both sides of the transgene (both sides), only on the PAM-proximal side (PAM proximal), or on the PAM-distal side (PAM distal). **b** Genomic and transgene sequences, 20 base pairs impaired homology sequences indicated in black, blue, and magenta; blue: gRNA target, magenta: PAM sequence, and green triangles highlight the cut location and gRNA-GFP transgene insertion site. GFP transgene is shown in green. Dashes highlight the missing sequences in the impaired homology constructs. The green (gRNA) and red (Cas9) markings on the y-axis in (**a**) indicate the average inheritance value of the control cross, with perfect homology, from Fig. 2a, e. **c–f** A schematic of each construct showing how the homology is impaired during the repair process. The bottom portion of the panel contains a 2D-correlation plot in which the inheritance of the DsRed-marked construct (Cas9) is represented on the x-axis, while the inheritance of the GFP-gRNA cassette is indicated on the y-axis. The individual data points are overlaid on a 2D-density distribution plot, and a 1D distribution of the data on each axis is also shown on the top and to he right of each graph. (**c**) Control: 2D-correlation graph of the data from Fig. 2 (perfect homology in both transgenes) representing the baseline. **d** Homology impaired on both sides of the gRNA transgene reduces allelic conversion of the GFP-marked construct. **e** Impaired homology on the PAM-proximal side behaves similarly to the control. **f** Impaired homology on the PAM-distal side seems to be slightly affected in terms of GFP inheritance, as individual data points are distributed almost equally on both sides of the diagonal. **c–e** Quantification of the percentage of data points falling over (green), on (black), and under (red) the midline is overlaid on each graph. Raw phenotypical scoring is provided as "Supplementary Data 4".

considered full-GD systems: (i) at an autosomal locus (full GD), and (ii) at an X-chromosome locus (full GDX). For the purpose of model exploration, we used ballpark parameters for each system: (i) a cleavage frequency of 100%, (ii) an allelic conversion efficiency of 50–100%, and (iii) no fitness costs associated with the Cas9 or gRNA alleles. All resistant alleles were assumed to be in-frame/cost-free. We modeled releases of Aedes aegypti, the mosquito vector of dengue, Zika, and chikungunya viruses, and simulated five weekly releases of 100 adult males homozygous for each system into a population with an equilibrium size of 10,000 adults. Model predictions were computed using 50 realizations of the stochastic implementation of the MGDrivE simulation framework[41].

Exploratory results for these parameter estimates suggested that the tGDc system, spread across two loci, performs very similarly to a full GD with some potentially beneficial qualities. At high allelic conversion efficiencies (90–100%), both systems spread at similar speeds; but as the allelic conversion efficiency declined (50–90%), the full GD spreads slightly quicker than the tGDc system (Fig. 6a, b). Resistant alleles accumulated to similar overall proportions for both systems (Fig. 6a), though because the tGDc system is spread across two loci, a higher proportion of individuals had at least one copy of a transgene at equilibrium (for allelic conversion efficiencies <100%) (Fig. 6c, d), with almost all individuals having at least one copy of a transgene at equilibrium for allelic conversion efficiencies of 90–100%. This could be advantageous for population replacement strategies, where a disease-refractory cassette could be linked to both the Cas9 and gRNA components of the tGDc system.

Within the tGD systems, having the components on autosomal loci seems to be the most effective design based on this exploratory modeling exercise. Autosomal systems spread faster

than X-linked systems due to their ability to drive in both sexes (Supplementary Fig. 8a, b). Interestingly, the linked tGD system spreads slightly faster than the unlinked tGDc system at moderate-to-low allelic conversion efficiencies (~50%), presumably due to the fact that, in the linked tGD, the elements are more often inherited together due to linkage, though this difference was modest or unnoticeable for higher allelic conversion frequencies (90–100%) (Supplementary Fig. 8c). Autosomal systems also result in a higher proportion of individuals with at least one copy of the transgene at equilibrium (Supplementary Fig. 8c, d). While these results are preliminary, neglecting fitness costs and detailed ecological considerations, they suggest potential benefits of the tGD system for population replacement in the field that warrant further investigation.

To further explore the potential of the tGD system in the field, preliminary modeling of a system intended for population suppression was studied in which the gRNA locus targets a gene required in at least one copy for female fertility. In this case, the tGD system behaves analogously to an equivalent full-drive approach intended for population suppression, but we envisioned some potential benefits in the tGD arrangement compared with the full GD (Supplementary Fig. 9). For an allelic conversion frequency of 100%, both the full GD and tGD systems induce a population crash within 1.5 years of the releases; however, the population quickly rebounds in both cases for an allelic conversion frequency of 99% or higher due to resistant alleles emerging at the female fertility locus that preserve fertility, thus conferring a selective advantage and preventing a crash (Supplementary Fig. 9). Tolerable rates of resistant allele generation are related to the inverse of the population size that one wishes to suppress[34]. One potential advantage of the tGD population suppression system is that a

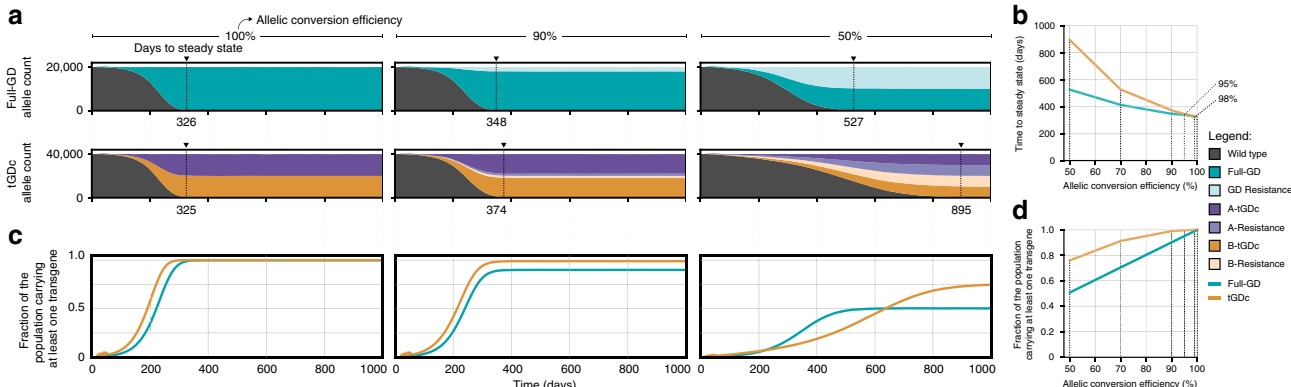

**Fig. 6 Comparison of full-drive and tGD spread in modeled populations.** Model predictions for releases of *Ae. aegypti* mosquitoes homozygous for the tGD and full-drive systems, parameterized with ballpark estimates: (i) a cleavage frequency of 100% in females and males, (ii) an allelic conversion efficiency, given cleavage, of 50–100% in females and males, and (iii) no fitness costs associated with the Cas9 or gRNA alleles. All resistant alleles are assumed to be in-frame/cost-free. Five weekly releases are simulated, consisting of 100 adult males homozygous for each system, into a population having an equilibrium size of 10,000 adults. Model predictions were computed using 50 realizations of the stochastic implementation of the MGDrivE simulation framework. **a** Stacked allele counts over time for the full-drive and tGD systems for allelic conversion efficiencies of 100, 90, and 50%. **b** Allelic conversion efficiency plotted against time to steady state for the full-drive (turquoise) and tGD (yellow) systems. **c** Fraction of the population carrying at least one transgene over time for the full-drive (turquoise) and tGD (yellow) systems for allelic conversion efficiencies of 100, 90, and 50%. **d** Allelic conversion efficiency plotted against the fraction of the population carrying at least one transgene at equilibrium for the full-drive (turquoise) and tGD (yellow) systems.

disease-refractory cassette could be linked to the Cas9 locus such that, in the event of resistant allele generation at the gRNA locus, the disease-refractory trait would still be propagated (Supplementary Fig. 9).

## Discussion

The tGD described herein splits the two genetic components required for a gene drive into two complementing elements located at separate chromosomal sites, which, when combined through genetic crossing, displayed super-Mendelian inheritance at three different loci (*e*, *y*, and *w*). We provide evidence for the advantages of a bipartite tGD system that allows flexible mix and match of gene-drive components to study key drive parameters and optimize the overall drive efficiency before any field application. As opposed to a full GD that propagates both Cas9 and gRNA as a single unit lacking intrinsic malleability to study their behavior, we show that the tGD is amenable to combinatorial optimization since one element can be independently modified to study its effect on the overall gene-drive efficiency. In addition, this aspect potentially reduces the number of transgenic lines needed for a given study.

Addressing one of the most pressing issues in the gene drive engineering community, our tGD approach also increases laboratory safety practices, since it greatly reduces potential spreading in the case of accidental escape of the laboratory animals, as the Cas9 and gRNA elements are kept as different lines that are combined only during experimentation or implementation[15].

Mathematical simulation suggests significant advantages of using tGD over full GD technologies for population modification. Due to its bipartite nature, the tGD can lead to a higher number of beneficial transgenes in a population at equilibrium (e.g., antimalarial gene)[42]. Indeed, our work paves the way for safer gene-drive research and provides a quicker and more systematic gene-drive optimization strategy to help move these technologies to mosquitoes and other insect pests.

## Methods

**Fly rearing and maintenance for experiments**. Fly stocks were raised at 18 °C with a 12/12-h day/night cycle on regular cornmeal medium. Experimental flies were grown at 25 °C with a 12/12-h day/night cycle. For TMP experiments, we used Formula 4-24 Instant Drosophila Food (Carolina Biological Supply Company, Cat. # 173214). After weighing 1 g of food per tube, we reconstituted it by adding 3 ml of water or water containing different concentrations of TMP (Oakwood Chemical, Cat. # 036441) dissolved in DMSO (Fisher Scientific, Cat. # D128). Flies were anesthetized using $CO_2$ to select individuals for crossing and phenotyping, and were scored by tracking fluorescent markers with a Leica M165 FC Stereo microscope with fluorescence. We used the DsRed and EGFP (referred to in the main text as GFP) markers as evidence for successful conversion. For the yellow body phenotype, we did not track mosaicism. For the eye phenotype, we scored white, red, or mosaic eyes (see Supplementary Data 1–4). All the work presented here followed procedures and protocols approved by the Institutional Biosafety Committee from the University of California San Diego, complying with all relevant ethical regulations for animal testing and research. Gene-drive experiments were performed in a high-security Arthropod Containment Level 2 (ACL2) barrier facility. For such experiments, shatterproof polypropylene plastic vials (Genesee Scientific Cat. # 32-113RL) were used and all gene-drive containing flies were disposed of by freezing for 48 h and subsequent autoclaving before being discarded as biohazardous material.

**Plasmid construction**. Standard molecular biology techniques were used to generate all constructs analyzed in this work. Constructs were built by Gibson assembly using NEBuilder HiFi DNA Assembly Master Mix (New England Biolabs, Cat. # E2621). Tables listing the PCR template and a pair of oligos used to amplify each DNA fragment used to build plasmids generated in this work can be found in the "Supplementary Methods" section in the Supplementary Information file. After assembly, plasmids were transformed into NEB 5-alpha Electrocompetent Competent *E. coli* (New England Biolabs, Cat. # C2989). Correct clones were subsequently confirmed by restriction analysis and Sanger sequencing. The final DNA sequence information for the constructs is available on NCBI; accession numbers are provided for each construct in the Supplementary Information file.

**Transgenic line generation and genotyping**. Constructs inserted at the *ebony*, *yellow*, and *white* loci were co-injected with a Cas9-expressing plasmid (pBS-Hsp70-Cas9 was a gift from Melissa Harrison & Kate O'Connor-Giles & Jill Wildonger [Addgene plasmid # 46294; http://n2t.net/addgene:46294; RRID: Addgene_46294]) and a pCFD3 plasmid (pCFD3-dU6:3gRNA was a gift from Simon Bullock [Addgene plasmid # 49410; http://n2t.net/addgene:49410; RRID: Addgene_49410])[43] expressing previously validated gRNA-*e1*[43], gRNA-*y1*[1,44], or gRNA-*w2*[44], respectively. Constructs used for transgenesis are outlined in Supplementary Fig. 1. We marked the Cas9 and gRNA constructs with either a DsRed (Red) or an EGFP (Green) fluorescent reporter expressed in the eye using the 3xP3 promoter. All injections to generate transgenic flies were performed by BestGene, Inc. or Rainbow Transgenic Flies, Inc. All constructs were injected into an isogenized Oregon-R (Or-R) strain from our laboratory to ensure a consistent background throughout all our experiments. After sending the constructs to the injection companies, we received 80–120 injected larvae. Once they hatched, we placed all G0 adults in different tubes (5–6 females crossed to 5–6 males). Then, G1 progeny were screened for positive flies with the fluorescent marker expressed in the eyes, which was indicative of the successful transgene insertion. Flies positive

for the marker were crossed individually to the same Or-R flies used for injection to make a homozygous stock in subsequent generations by identifying the *e*, *y*, or *w* visible marker. Last, we sequenced each stock to confirm correct transgene integration.

**Molecular analysis of resistant alleles**. For resistant allele sequence analysis, we performed single-fly DNA extractions following the protocol described by Gloor GB and colleagues[45]. We added 200 μL of water to dilute each sample to a final volume of 250 μL and used 1–5 μL of each DNA extraction as a template for a 25-μL PCR reaction. We performed PCRs covering the gRNA cute site for either the *yellow* or *white* locus in order to sequence the resistant allele present. The oligos used can be found in the Supplementary Information.

**Graph generation and statistical analysis**. We used GraphPad Prism 7 to generate all our graphs. For statistical analysis, we used the Statkey analysis tool [http://www.lock5stat.com/StatKey/index.html]. We performed a Randomization Test for a Difference in Means when comparing our experimental conditions (Supplementary Data 1, Supplementary Data 2 and Supplementary Data 4). In Fig. 5 we also performed a Randomization Test for a Difference in Proportions (Supplementary Data 4) to evaluate differences in the distribution of the fraction of data points below the diagonal. In both cases we have performed 5000 randomizations of our data.

**Mathematical modeling**. To model the expected performance of the transcomplementing gene-drive system in populations of *Aedes aegypti*, the mosquito vector of dengue, chikungunya, and Zika viruses, we simulated release schemes for the transcomplementing system with: (i) components linked on an autosome (tGD), (ii) components unlinked at two autosomal loci (tGDc), (iii) components linked on the X chromosome (tGDX), and (iv) components unlinked at two loci on the X chromosome (tGDXc). We also compared the system with standard full gene drives at an autosomal locus (Full-GD), and at an X-chromosome locus (Full-GDX). Releases were simulated consisting of 5 weekly releases of 100 adult males homozygous for each system using the MGDrivE simulation framework[41] [https://marshlllab.github.io/MGDrivE/]. This framework models the egg, larval, pupal, and adult mosquito life stages (both female and male adults are modeled) implementing a daily time step, overlapping generations, and a mating structure in which adult males mate throughout their lifetime, while adult females mate once upon emergence, retaining the genetic material of the adult male with whom they mate for the duration of their adult lifespan. Density-independent mortality rates for the juvenile life stages are assumed to be identical and are chosen for consistency with the population growth rate in the absence of density-dependent mortality. Additional density-dependent mortality occurs at the larval stage, the form of which is taken from previous studies[46]. The inheritance patterns for the tGD, tGDc, tGDX, tGDXc, Full-GD, and Full-GDX systems are modeled within the inheritance module of the MGDrivE framework[41]. We parameterized our transcomplementing and full gene-drive models using ballpark parameter estimates for model exploration: (i) a cleavage frequency of 100% in females and males, (ii) a frequency of accurate homology-directed repair, given cleavage, of 50–100% in females and males, (iii) no fitness costs associated with the Cas9 or gRNA alleles, and (iv) all resistant alleles being in-frame/cost-free. We implemented the stochastic version of the MGDrivE framework to capture the randomness associated with low genotype frequencies and rare events such as resistant allele generation under some parameterizations. The code for running the simulation is freely available from the MGDrivE GitHub repository [https://github.com/MarshallLab/MGDrivE], and the package can be installed on R through CRAN [https://cran.r-project.org/web/packages/MGDrivE/]. The inheritance cubes used in these simulations are the "cubeTGD" and "cubeTGDX" variants of the codebase. Parameter values used in *Aedes aegypti* population model are reported here below:

$\beta$: Egg production per female $(day^{-1})$[47]—value: 20
$T_E$: Duration of egg stage $(days)$[48]—value: 5
$T_L$: Duration of larval stage $(days)$[48]—value: 6
$T_P$: Duration of pupal stage $(days)$[48]—value: 4
$r$: Daily population growth rate $(day^{-1})$[49]—value: 1.175
$\mu_M$: Daily mortality rate of adult stage $(day^{-1})$[50,51]—value: 0.090
$N$: Adult female population size[52]—value: 10,000

**Reporting summary**. Further information on research design is available in the Nature Research Reporting Summary linked to this article.

## Data availability
The sequence of all plasmid constructs generated in this paper has been deposited into the GenBank database with accession codes:
MN551085[https://www.ncbi.nlm.nih.gov/nuccore/MN551085],
MN551086[https://www.ncbi.nlm.nih.gov/nuccore/MN551086], MN551087[https://www.ncbi.nlm.nih.gov/nuccore/MN551087], MN551088[https://www.ncbi.nlm.nih.gov/nuccore/MN551088], MN551089[https://www.ncbi.nlm.nih.gov/nuccore/MN551089], MN551090[https://www.ncbi.nlm.nih.gov/nuccore/MN551090], MN551091[https://www.ncbi.nlm.nih.gov/nuccore/MN551091], MN551092[https://www.ncbi.nlm.nih.gov/

nuccore/MN551092], MN551093[https://www.ncbi.nlm.nih.gov/nuccore/MN551093], MN551094[https://www.ncbi.nlm.nih.gov/nuccore/MN551094].

In this study, we accessed the Addgene plasmid # 49411 [http://n2t.net/addgene:49411]; RRID:Addgene_49411. All raw phenotypical scoring data collected are reported in Supplementary Datas 1–4 files in Microsoft Excel format (.xlsx). All other data are available from the authors.

## Code availability
The code for running the simulation of the mathematical modeling presented in the paper is freely available from the MGDrivE GitHub repository [https://github.com/MarshallLab/MGDrivE], and the package can be installed on R through CRAN [https://cran.r-project.org/web/packages/MGDrivE/]. The inheritance cubes used in the simulations are the "cubeTGD" and "cubeTGDX" variants of the codebase.

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

## Acknowledgements

We thank Bill McGinnis, Steve Wasserman, Mike Perry, Kaycie Butler, and members of the Gantz laboratory for comments and edits on the paper. We thank Emily Bulger and Shannon Xu for experimental contribution on generating reagents. Research reported in this paper was supported by the University of California, San Diego, Department of Biological Sciences, by the Office of the Director of the National Institutes of Health under award number DP5OD023098 and by a DARPA Safe Genes Program Grant (Brdi N66001-17-2-4055). A Paul G. Allen Frontiers Group Distinguished Investigators Award supported E.B., a gift from the Tata Trusts of India to TIGS-UCSD supported X.F., a DARPA Safe Genes Program Grant (HR0011-17-2-0047) supported J.B. and J.M.M., and funds from the UC Irvine Malaria Initiative supported H.M.S.C. and J.M.M. The fruit fly drawing used in Fig. 4 was adapted from the original artwork by Madboy74 [CC BY-SA 4.0] [https://creativecommons.org/licenses/by-sa/4.0/].

## Author contributions

E.B. and V.M.G. conceived the project. V.L.D.A. and V.M.G. contributed to the design of the experiments. V.L.D.A., V.M.G., A.L.B. and X.F. performed the experiments and contributed to the collection and analysis of data. H.M.S.C., J.B. and J.M.M. designed and performed the mathematical modeling experiments. V.L.D.A., J.M.M. and V.M.G. wrote the paper. All authors edited the paper.

## Competing interests

V.M.G. and E.B. have equity interests in Synbal, Inc. and Agragene, Inc., companies that may potentially benefit from the research results and also serve on the company's Scientific Advisory Board and Board of Directors. The terms of this arrangement have been reviewed and approved by the University of California, San Diego in accordance with its conflict of interest policies. V.L.D.A., A.L.B., X.F., H.M.S.C., J.B. and J.M.M. declare no competing interests.
