## [Peer Review File · Nature Communications]

Reviewers' Comments:

Reviewer #1:

Remarks to the Author:

In this study, the authors developed homing gene drives in which the gRNA and Cas9 elements were separated. The gRNA element, however, provided gRNAs that enabled drive of both elements. This was referred to as a "trans-complementing gene drive" (tGD). The authors demonstrate that their tGDs work efficiently and investigate formation of resistance alleles. They then use a small molecular to activate a special version of Cas9 to control drive activity. Next, they investigate the effects of impaired homology on homing efficiency, and finally, they model the performance of tGDs.

Overall, the study can probably be made scientifically sound with some revisions, and it has some interesting aspects. However, much of the study is a repeat of previous work from several studies, and the remainder is unlikely to be of high impact alone. Details are below. Thus, I would have reservations about publishing the work in Nature Communications even after revisions. Here are a few questions, comments, and ways that the manuscript could be improved. Page numbers include the title page.

1. Abstract and introduction: Advertising tGDs as a safe alternative does not seem correct to me. While individual elements can be maintained in the long term in a safe manner if isolated, they could still mix if accidentally released together. More importantly, they are not safe against accidental release when being tested, which is the most likely time that an accidental release would occur in the first place. Thus, I would consider any tGD safety benefits to be minor, worth at most a single sentence in the discussion, rather than being featured as a primary selling point of the manuscript.

The authors are still overall in compliance with safety guidelines because they used a containment facility and targeted genes that are critical for fly fitness in the wild using X-linked constructs that have substantial resistance due to maternally deposited Cas9.

2. Introduction: Aside from safety, which doesn't seem to be as useful as advertised, are there any other "selling points" for tGD compared to normal complete homing drives? Maybe the higher conversion efficiency (though this is unclear)?

3. Introduction: The idea of splitting up critical components of the drive is not novel. The authors properly acknowledge the split drives previously, but no attention is given to "integral drives" (<https://bio.biologists.org/content/8/1/bio037762>), which use a similar method of splitting the drive into two components (in this case, payload and drive elements). These drives have been considered theoretically in the link and have already also been successfully tested in mosquitoes (though this experimental work has yet been published).

4. Introduction, paragraph 3: while the safety issue was given time in the introduction (though see #1), the reasons for investigating the other aspects of this manuscript are unclear. A more extensive treatment may be helpful.

5. Page 4-5: The study on ebony doesn't seem to offer anything compared to the next study other than a different target site. It probably belongs in the supplement. Why didn't the authors assess drive conversion in males for the construct at the ebony locus? It seems easy to do for completely, and it would be modestly useful to have another point for comparison of male vs. female drive efficiency in *Drosophila*.

6. Page 6: there doesn't seem to be any statistical comparisons between the different inheritance

values, so it's hard to say which differences may be real and thus, which comparisons and changes may be interesting.

7. Page 6: In reference 21, it seems that a small driving element may have higher drive efficiency than a "standard" drive, which would also apply to this study, though exact comparisons may not be possible in this or the authors own study (could be worth considering, though). This previous study doesn't seem to be acknowledged when the authors report higher conversion in their own drives in page 6, line 127. Also, previous standard drives at both white and yellow (Champer references 2 and 17) had lower drive conversion, so size could easily still be an issue. In both the authors' tests flipping the location of the Cas9 and gRNAs, the larger Cas9 element appeared to have lower efficiency, though lack of statistical comparisons makes this unclear. It is likely that the effects in this study and others involve a combination of factors such as different expression or gRNA activity at different loci, making them tricky to disentangle.

8. Page 8, line 157: please check references. The correct references should probably be 2,4,5,17,21. Also, the authors are trying to draw a contrast with previous full GD work when none exists. In almost all previous Drosophila and mosquito studies, Cas9/gRNA was often maternally deposited and prevented drive conversion in the next generation. This was also the case in this study, so it is unclear why the authors say it is not. If tGDs were released in the wild for purposes of altering a population, maternal effects would be just as important as with a regular gene drive.

9. Page 8, lines 158-169: the fact that both Cas9 and gRNA must be maternally deposited to form resistance alleles in the early embryo was already shown in Champer et al. reference 21. The use of this reference, to encourage new gene drive designs, is not sufficient to show this.

10. Page 8, lines 170-182: again, nanos has been well-studied in homing drives. All of the Champer references use nanos, and reference 17 directly compares it to vasa.

11. Page 9, lines 202-212: are these from crossing scheme D (the one with ~50% inheritance as noted)? If so, references 2 and 17 already discovered this same thing for yellow and white loci embryo resistance alleles by direct sequencing. It's still nice to see larger sample sizes, though. If the topic is germline resistance formation, then for a more precise analysis of germline resistance allele formation timing, see reference 17 and this other paper that is not referenced (<https://www.pnas.org/content/115/24/6189>)

12. Figure 3 C-J: I think the take-home message for these will be lost for most readers. Additionally, the format could be improved. Maybe bars of variable height for different numbers of sequences, with the bars divided based on how many unique sequences there are?

13. Figure 4A: maybe note that the mosaicism was due to leaky somatic expression from vasa-Cas9? This should be clear comparing to nanos, which shouldn't have such expression in significant quantity.

14. A general comment at this point: there doesn't seem to be any direct measurements of reporting of the germline and embryo resistance allele formation rates in these construct, even though the necessary data seems to be present.

15. The small molecule control method (the same method from partly the same group) was already introduced in this other manuscript (<https://www.biorxiv.org/content/10.1101/665620v1>). It's fine to include in the authors' manuscript, but it doesn't show anything truly different compared to the previous manuscript, other than a brief test in tGDs.

16. The concept of impaired homology reducing drive efficiency is not novel or unexpected and was first measured in reference 17 and examined in more detail (with modeling and additional experiments) in the following bioRxiv reference (<https://www.biorxiv.org/content/10.1101/679902v1>). These indicated that impaired homology on even one side of the cut was enough to reduce drive conversion efficiency. The authors may not have seen this due to the small size of their impaired homology. The interesting part of this section is that doubly impaired homology seems to more substantially impair drive conversion than predicted by a multiplicative combination of impaired homology on either side. It is weird that it seemed to affect the Cas9 element too a bit (maybe reduced gRNA expression), but the lack of statistical comparisons makes one pause before drawing meaningful conclusions.

17. Page 15, lines 372-374: would this provide any benefit over simply using two complete gene drive alleles located at different loci, each with the payload? Since the proposed system has two components and would thus be nearly as difficult to construct as two separate drives, the authors need to show superiority to this method if they want to highlight their tGD as being an interesting alternative.

18. Methods: unfortunately, the methods for plasmid construction are unacceptable. While it is okay to promise full sequences on NCBI at a future date, this makes it impossible to review. Furthermore, the manuscript in any case would need primer sequences and plasmid construction details, plus substantially more information on "standard molecular biology techniques."

19. Methods: the specific code for the MGDive simulations should be made available.

Minor issues:

20. Abstract: "bypassing the dictates of Mendelian genetics" is clear for people who understand gene drive, but it won't help people who are new to the field learn what it is.

21. Introduction, paragraph 2: I don't think it's accurate to say that only two approaches are "feasible". After all, the authors' tGD approach was certain "feasible" before they developed their constructs.

22. Page 6, line 125: "alleviating" isn't an accurate word to use here (or as used elsewhere in the manuscript). I'm not exactly sure what the authors are trying to convey.

23. Page 6, line 128: w118 should be w1118.

24. Page 7, line 145-146: cite the other studies or refer to only one study.

25. Page 7, line 148: pathway should be pathways, though this phrase is a bit awkward in the first place. Maybe delete the word "one"?

26. Page 8, line 152: references 2 and 17 specifically showed this.

27. Page 8, line 156: "in" should perhaps be "is".

28. Page 8, lines 185-190: it is unclear if the authors are looking at resistance alleles to analyze resistance alleles formed from maternal effects or resistance alleles formed in the germline of their F1 females. It appears to be the latter, but the paragraph starts out discussing the former.

29. Page 9, line 201: what is quality here?

30. Page 9, line 204: If it looks like four it could be a lot more than four. There are only four types of nucleotides, making it appear that this is the max...

31. Figure 5A: maybe include the drive without impaired homology for comparison? It would be repeating previously shown data, but the comparison could be worth it. Alternatively, just indicate what this level is.

32. Page 16, lines 378-380: this is presumably because the two components will be together more often despite imperfect conversion rates.

33. Supplementary Figure 1: the term "short HAs" is not a very good description of the homology arms here. It would conjure an image of a HA on a plasmid that is just shorter than the usual ~1kB. Instead, the authors may want to use "truncated HA" or something similar that implies that the HAs do not extend to the cut site.

Overall, the authors should substantially revise their manuscript to incorporate previous work into their discussion and clarify several issues. Much of the manuscript can be presented as useful confirmatory results, but the manuscript does need to have many of the above revisions implemented to be scientifically sound and ready for publication.

Reviewer #2:

Remarks to the Author:

This manuscript addresses several issues that are central to gene drive development and potential deployment including safe propagation, spatially/temporally restricted activity, resistant allele generation and use of multiple gRNAs. In general the experiments are clearly presented and the text is easy to follow. The figures are also well constructed although the text in some sections is very small and is illegible on a standard print version.

The (trans) drive design has not been published previously and is an interesting concept – a significant advantage of this approach is that the drive components can be individually propagated in complete safety. Further, multiple CAS9 promoters and gRNA targets can be readily combined and compared side-by-side for allele conversion and indel-generating activity. Additionally, as far as I am aware, this is the first demonstration of multiple gene conversion events in a single individual. Although not explicitly mentioned, the data have implications for "Daisy Drive" strategies which are predicated on multiple "simultaneous" homing events within an individual.

Comments

Fig. 1, 2 & 3: Thorough demonstration of the concept. One difference between Fig 1A and 2A that is not discussed is that the homing events in the latter occur on the same chromosome which may impact homing efficiency. Notably this is higher in 2A (same chromosome) than 1A (different chromosome). Can the authors speculate about this? It would be interesting to also generate the yellow/ebony drive (Fig. 1) using Nanos promoter – this would also help to address test whether allele conversion on the same chromosome is more efficient. Also, simultaneous cutting could result in chromosome translocation or inversion – was this investigated?

One experiment that I would really like to see is how the tGD system behaves in the male germline. The targeting of yellow in all experiments precludes this because this locus is X linked. As shown in

the modeling, the tGD works best when homing occurs in both sexes – however, only females are tested in the manuscript. Therefore I believe that it is also important to test the male germline.

Is the nanos promoter Cas9 construct smaller than Vasa-Cas9? This may also impact the efficiency of allele conversion.

Fig. 3./Supp Fig 4: It appears that no F2 progeny with WT sequence were identified from the sequencing analysis – please confirm.

Model in Fig 4 is plausible.

Figure 5. I found this experiment confusing. I disagree with the statement:

“However, as it is intrinsically impossible to have perfect homology to all DNA ends generated when using multiple gRNAs,...”. Cleavage of the WT chromosome at flanking sites will still result in perfect homology to the gene drive bearing chromosome – although the homology will not “but up” against the insertion. As I understand it, the experiment does not examine this scenario – rather it tests the impact of a lack of homology flanking the gene drive sequence. I cant see how these experiments relate to the multi-gRNA scenario.

Fig. 6: Preliminary modeling is included – useful addition to the manuscript. Some modeling analysis of how the trans system behaves for population suppression (rather than replacement) would be valuable.

Minor issues

“...alleviating the significant difference between the two transgenes that was observed for tGD(y,w) (89% at yellow and 96% at white)...” Significant difference implies a statistical analysis was performed. This does not appear to be the case.

Fig 3. The “Flies sequenced” numbers are very small and difficult to read.

Fig. 4B correct “zigote” (zygote)

Supp Fig 6 B' Cas9 versus DD2-Cas9 experiments should be labeled.

We are grateful for the Reviewers' assessment, which helped us to improve the manuscript substantially. We hope we have addressed all the concerns raised and answered each point to the best of our abilities. We are providing a revised version of the manuscript, including the suggested edits and revisions, with comments highlighting the edits addressing each specific comment.

Reviewers' comments:

Reviewer #1 (Remarks to the Author):

In this study, the authors developed homing gene drives in which the gRNA and Cas9 elements were separated. The gRNA element, however, provided gRNAs that enabled drive of both elements. This was referred to as a "trans-complementing gene drive" (tGD). The authors demonstrate that their tGDs work efficiently and investigate formation of resistance alleles. They then use a small molecule to activate a special version of Cas9 to control drive activity. Next, they investigate the effects of impaired homology on homing efficiency, and finally, they model the performance of tGDs.

Overall, the study can probably be made scientifically sound with some revisions, and it has some interesting aspects. However, much of the study is a repeat of previous work from several studies, and the remainder is unlikely to be of high impact alone. Details are below. Thus, I would have reservations about publishing the work in Nature Communications even after revisions. Here are a few questions, comments, and ways that the manuscript could be improved. Page numbers include the title page.

1. Abstract and introduction: Advertising tGDs as a safe alternative does not seem correct to me. While individual elements can be maintained in the long term in a safe manner if isolated, they could still mix if accidentally released together. More importantly, they are not safe against accidental release when being tested, which is the most likely time that an accidental release would occur in the first place. Thus, I would consider any tGD safety benefits to be minor, worth at most a single sentence in the discussion, rather than being featured as a primary selling point of the manuscript.

The authors are still overall in compliance with safety guidelines because they used a containment facility and targeted genes that are critical for fly fitness in the wild using X-linked constructs that have substantial resistance due to maternally deposited Cas9.

(1) We advertise the tGD as "safer" than a full-GD since during non-experimental windows the chances of releasing individuals capable of full-GD is dramatically reduced. Additionally, tGD allows for combinatorial testing of Cas9 and gRNA constructs, allowing the generation and maintenance of less lines. We elucidate our reasoning in the table below with a side-by-side

comparison involving the generation and testing of 12 gene drives in a 1-year time frame and amplifying and testing them over a period of 3 months:

Full-GD	tGD	Full-GD RISK	tGD RISK
12 transgenic lines need to be generated and maintained over a period of 1 year. (e.g.: 3x Cas9 genes and 4x gRNAs genes assembled combinatorially)	7 transgenic lines need to be generated and maintained over a period of 1 year. (e.g.: 3x Cas9 and 4x gRNAs lines)	More lines need to be maintained. Any 1 fly escaping has the potential to mate with a wild counterpart and spread in the population.	Less lines maintained lowers risk. 2 flies need to escape in a relatively close time window (2-4 weeks), AND they need to be a male and a female, AND need to be from a Cas9 line and a gRNA line AND they need to mate (either before leaving the laboratory or after).
12 lines need to be expanded for performing the experiments over a period of 1 month.	7 lines need to be expanded for performing the experiments over a period of 1 month.	Equal as above, 12 lines need to be amplified for experimentation.	Equal as above, 7 lines need to be amplified for experimentation.
12 experiments are run in parallel to evaluate the constructs over a period of 3 months.	12 experiments are run in parallel to evaluate the constructs over a period of 3 months.	Risk associated to the escape of an individual.	Risk comparable to a full-GD

Throughout the text we have used the term “safer” when comparing the tGD to a full gene drive and not “safe” per se as the tGD would display the similar exponential spread dynamics of a full-GD when combined; to be consistent with this point we have revised the title to:

*“Trans-complementing split-gene drive system provides flexible application for **safer** laboratory investigation and potential field deployment”*

2. Introduction: Aside from safety, which doesn’t seem to be as useful as advertised, are there any other “selling points” for tGD compared to normal complete homing drives? Maybe the higher conversion efficiency (though this is unclear)?

(2) We have three main “selling points” with associated benefits for the tGD technology that we are highlighting in the third paragraph of the introduction section.:

- The tGD generates simultaneous super-Mendelian inheritance of two separate, yet interdependent genetic elements. This has not been previously shown experimentally in other systems and these findings have implications for other strategies using multiple elements driving simultaneously such as the proposed “daisy-chain drive”, therefore addressing some needs of the CRISPR gene drive field (**novel**).
- As illustrated in the table above, the tGD allows for combinatorial optimization of gene drives, without building full gene drives (**safer**) as well as potentially building less transgenic lines to test the same conditions (**saves money and time**). Also, in the event tGD organisms (e.g., mosquitoes) were to be amplified in numbers for a field release, the two component stocks could be amplified separately and only crossed at the time of intended release. This would greatly reduce the likelihood of a premature escape of a driving insect.
- The bipartite nature of the tGD system makes it possible to spread a greater number of effectors in the field or to achieve deeper penetration of effectors carried in common by both component elements. Our modeling seems to display potential advantages discussed in the manuscript (**potentially beneficial in the field**).

We have added modifications in the text to better highlight these points.

3. Introduction: The idea of splitting up critical components of the drive is not novel. The authors properly acknowledge the split drives previously, but no attention is given to “integral drives” (<https://bio.biologists.org/content/8/1/bio037762>), which use a similar method of splitting the drive into two components (in this case, payload and drive elements). These drives have been considered theoretically in the link and have already also been successfully tested in mosquitoes (though this experimental work has yet been published).

(3) We agree with Reviewer#1 that the idea of splitting the critical components of a full gene drive is not novel as some of us have proposed it in 2015 (Gantz and Bier 2015, Bioessays, Ref.

19). This manuscript is a follow-up work describing the first experimental validation of our original concept publication.

We thank the reviewer for the suggestion of the Nash et al. 2019 which proposes theoretically an interesting integral gene drive strategy. In Nash et al. it seems to us that the effector cassette is split from the drive component, containing its own gRNA. Although, in this system the Drive component still retains both the Cas9 and gRNA elements together, and for this reason we would not consider this as a split gene-drive system comparable to either a gRNA-only or a tGD drive (Cas9 and gRNA in different individuals).

Here below the integral gene drive image from Nash et. al as a reference:

It seems to us that the integral gene drive strategy reported in Nash et al. would be more comparable to a full gene drive (Cas9 and gRNA in the same cassette) combined with a second (or third) gRNA at a second (or third) location (which also is able to spread) in the same animal. Since this theoretical system would allow for the simultaneous spread of two (or three) elements we added this reference in the first part of our results section where we believe it is more appropriate:

“The above results demonstrate for the first time that a CRISPR gene drive can be split into two separate genetic elements located on different chromosomes, which once combined, can be simultaneously propagated with super-Mendelian inheritance. This conditional property offers flexibility and increases safety while functioning as a full-GD. Furthermore, these findings have implications for other strategies that similarly use multiple elements driving simultaneously such as the proposed “daisy-chain drive”²⁴ or integral gene drives²⁵”.

4. Introduction, paragraph 3: while the safety issue was given time in the introduction (though see #1), the reasons for investigating the other aspects of this manuscript are unclear. A more extensive treatment may be helpful.

(4) We have revised this portion of the introduction to clarify our reasoning to investigate specific aspects that have been discussed in the field.

5. Page 4-5: The study on *ebony* doesn't seem to offer anything compared to the next study other than a different target site. It probably belongs in the supplement. Why didn't the authors assess drive conversion in males for the construct at the *ebony* locus? It seems easy to do for completely, and it would be modestly useful to have another point for comparison of male vs. female drive efficiency in *Drosophila*.

(5) We thank the reviewer for this comment. Our reason to keep the *ebony* data as a main figure is due to the fact that we believe this data helps the reader understand how the tGD system would perform at different genomic location (gRNAs in *ebony* vs. *white*) and when the elements are acting on different chromosomes (tGD(y,e) X and III chromosome vs. tGD(y,w) both on the X chromosome). We believe that the comparison is important as the same Cas9 transgene propagates more efficiently when the gRNA is transcribed from the *white* locus than when from *ebony* gene.

At first, we decided not to perform the *ebony* male germline study as in males it would technically not be a tGD system. Males have only one X chromosome and therefore only the construct in *ebony* would copy, while the construct in *yellow* would act as a regular transgenic source of Cas9. Upon this suggestion, however, we performed the experiment and added to the supplementary information section, and modified the text accordingly. Since the genetic situation would not be fully comparable to the tGD for the reason stated above, we maintained this experiment as a minor point in the manuscript.

6. Page 6: there doesn't seem to be any statistical comparisons between the different inheritance values, so it's hard to say which differences may be real and thus, which comparisons and changes may be interesting.

(6) We have performed a two-tail randomization test for difference in means and calculated the p-value and added it to the **Supplementary Data 2** for relevant comparisons and modified the text accordingly.

7. Page 6: In reference 21, it seems that a small driving element may have higher drive efficiency than a "standard" drive, which would also apply to this study, though exact comparisons may not be possible in this or the authors own study (could be worth considering, though). This previous study doesn't seem to be acknowledged when the authors report higher conversion in their own drives in page 6, line 127. Also, previous standard drives at both *white* and *yellow* (Champer references 2 and 17) had lower drive conversion, so size could easily still be an issue. In both the authors' tests flipping the location of the Cas9 and gRNAs, the larger Cas9 element appeared to have lower efficiency, though lack of statistical comparisons makes this unclear. It is likely that the effects in this study and others involve a combination of factors such as different expression or gRNA activity at different loci, making them tricky to disentangle.

(7) We agree with the Reviewer that the size of the constructs could influence the conversion efficiency. Although some of our data are controversial on this aspect:

1. When analyzing our swapped tGD(w,y) construct data (**Supplementary Figure 2**), it seems that at the *yellow* location the smaller gRNA construct gets copied more efficiently (95%) than the Cas9 of our tGD(y,w) (89%; **Fig. 2A,E**) (p-value < 0.0004 see **Supplementary Data 2**). As Reviewer #1 points out could be due to the difference in size.
2. When we instead look at the *white* locus we see that the larger Cas9 in the swapped tGD(w,y) construct (**Supplementary Figure. 2**) is copied also at a significantly higher rate than the smaller gRNA one in our tGD(y,w) arrangement (**Fig.2A,E**): 98% in tGD(w,y) vs 96% in tGD(y,w) (p-value = 0.038). This suggests that, at least in this case, an increase in construct size seems to increase efficiency.
3. When comparing copying of the *vasa*-Cas9 construct in the tGD(y,e) (**Fig.1**) with the tGD(y,w) (**Fig.2A,E**) we observe a difference 83% vs 89% (p-value = 0.0084) suggesting that **the location of gRNA expression seems to influence the conversion efficiency**, which therefore makes questionable any conclusion correlating copying efficiency and size in our swap experiment.

For these reasons, we decided not to make any conclusion on the construct size's impact on efficiency as we do not believe any data in our work or in comparison with published work would be enough to do so.

As Reviewer #1 suggested we have added these statistical comparisons in the **Supplementary Data 2** relevant to these points and modified the text accordingly.

Lastly, we have added Reference 21 at the location suggested by Reviewer #1.

8. Page 8, line 157: please check references. The correct references should probably be 2,4,5,17,21. Also, the authors are trying to draw a contrast with previous full GD work when none exists. In almost all previous *Drosophila* and mosquito studies, Cas9/gRNA was often maternally deposited and prevented drive conversion in the next generation. This was also the case in this study, so it is unclear why the authors say it is not. If tGDs were released in the wild for purposes of altering a population, maternal effects would be just as important as with a regular gene drive.

(8) We thank the reviewer for catching this mistake. We have replaced references with 2,4,5,17 in line 157 and kept ref 21 separately discussed in the following paragraph.

9. Page 8, lines 158-169: the fact that both Cas9 and gRNA must be maternally deposited to form resistance alleles in the early embryo was already shown in Champer et al. reference 21. The use of this reference, to encourage new gene drive designs, is not sufficient to show this.

(9) We thank the reviewer for the comment, following the suggestion we moved Reference #21 from line 157 to the previous paragraph where we believe it is more appropriate.

We would like to point out that in our work we are using the same system tGD(y,w) to analyze two different situations:

- 1) The two elements (Cas9 and gRNA) are inherited **separately** from either F0 parent (Fig. 2A,B,E)

In Champer et al. (Ref. 21) it was shown that in a gRNA-only drive (split drive) there is no maternal effect when the two elements (Cas9 and gRNA) are inherited separately, analogously to what we show in **Fig. 2A,B,E**.

- 2) The two elements (Cas9 and gRNA) are inherited **together** from either F0 parent (**Fig. 2C,D,E**)

In Champer et al. (Ref. 21), we could not find an analogous experiment in which the same Cas9 and the gRNA-only drive constructs are co-inherited from the F0 parent. Other experiments in Ref. 21 use the split system to explore the effects of maternally deposited Cas9 protein (but not genetically inherited) differently from what we performed in this work.

We believe that our work includes a well-controlled analysis of the maternal effect due to deposition of Cas9 and gRNAs in the egg using the tGD(y,w) system. Our data expands on the work in Ref. 21 by analyzing in detail the types of indels generated in each genetic situation.

10. Page 8, lines 170-182: again, nanos has been well-studied in homing drives. All of the Champer references use nanos, and reference 17 directly compares it to vasa.

(10) We believe that our study provides a different analysis of the *vasa* and *nanos* Cas9 drivers. In our work the two constructs have been inserted at the same genomic location and, differently from Champer et al. (Ref 17), they vary only in the Cas9 regulatory region (*vasa* or *nanos*), maintaining the same Cas9 coding sequence and nuclear localization sequences (NLS) used, factors that have been shown to affect editing efficiency in a Cas9-based base editor system (Koblan et al. 2018, Nat. Biotech. - <https://www.nature.com/articles/nbt.4172>) stating:

"These findings establish that improvements in nuclear localization and codon usage that benefit BE4 also enhance ABE efficiency."

Our *vasa* vs. *nanos* comparison differs on these points:

- **Cas9 DNA coding sequence** - In Champer et al. (Ref 17) the Cas9 sequence has different optimization codons used in the two constructs (*nanos* vs *vasa*) which could affect functionality.
- **Nuclear Localization sequences (NLS) used** - Additionally in Champer et al. (Ref 17) the *nanos* Cas9 has only 1x NLS instead of 2x used in the *vasa* constructs, these factors could additionally affect functionality between the constructs previously analyzed.

Additionally, since tGD allows us to analyze the copying efficiency at separate locations (Cas9 and gRNAs) our data allows us to perform a controlled analysis of the two drivers (*vasa* and *nanos*) acting on the same exact drive element (gRNAs construct in *white*). It is true that this comparison has been previously reported in the literature (*nanos* vs. *vasa*), and while we decided to keep the *nanos* data as a supplement, our data show a different behavior of the two drivers when compared to previous work. This information could be of help for the gene drive field or other researchers using the same constructs.

11. Page 9, lines 202-212: are these from crossing scheme D (the one with ~50% inheritance as noted)? If so, references 2 and 17 already discovered this same thing for yellow and white loci embryo resistance alleles by direct sequencing. It's still nice to see larger sample sizes, though. If the topic is germline resistance formation, then for a more precise analysis of germline resistance allele formation timing, see reference 17 and this other paper that is not referenced (<https://www.pnas.org/content/115/24/6189>)

(11) We thank the reviewer for the comment. The cross schemes refer to experiments in which Cas9 and gRNAs were inherited together from F1 females carrying both paternal (**Fig. 2C**) and maternal (**Fig.2D**) drive alleles and with both *vasa* (**Fig. 2**) and *nanos* (**Suppl. Fig. 3**) promoters.

We would like to clarify that we estimated the diversity of resistance alleles in the germline of **single F1 females (Fig.2C-D)** by extensively sampling and sequencing resistant alleles in F2 males. We modified the text accordingly to better clarify this in the manuscript.

Regarding References 2 and 17, we have found some differences between our work and the mentioned references:

1. In reference 2, the authors analyzed two full gene drive scenarios targeting the *yellow* gene and driven by both *vasa* or *nanos* promoters. Resistant alleles occurring at the *yellow* locus were analyzed performing Sanger sequencing (analogously to what we did), but these samples seem to come from genetic crosses performed in batch as stated in the manuscript:

*“While the alleles originating from the vasa construct appeared randomly distributed, possibly due to lower sample size per parent, four out of six of the alleles from the nanos construct that were found in more than one fly were found in flies that shared the same batch of parents (**composed of 2–4 females**). These data support the idea that resistance alleles in the nanos drive could have formed in early germline stem cells that eventually gave rise to multiple progeny, though it does not rule out the formation of resistance alleles by other pathways as well.”*

In our study we show that specific indels occur at much higher frequencies than others (**Fig. 3A,B**). Therefore, the progeny of a pooled cross could carry the same indel generated in the germline of different mothers, making it harder to draw conclusions from such analysis.

- In our manuscript all crosses were performed in single-pairs allowing us to analyze in depth the types of indels generated in each single F1 germline by sequencing F2 males, a task that could not have been achieved with the same accuracy using pooled crosses.
2. Second, in reference 17 the authors present two full gene drives using one gRNA and using *vasa* and *nanos* promoters, in this case they instead targeted the *white* gene.

As we understand, in reference 17, F1 females carrying the drive element (with either maternal or paternal drive from the F0) were crossed in single-pairs to wild-type males to assess inheritance of the gene drive in the F2 progeny. In this reference the authors state that:

“Sequencing of resistance alleles from our one-gRNA nanos and vasa drives targeting white (SI Appendix, SI Results) revealed that flies from the same mother were significantly more likely to have a sibling with an identical resistance allele compared with an unrelated fly ($P < 0.0001$ for nanos and $P = 0.0021$ for vasa, paired t test). This suggests a mechanism in which at least a significant fraction of germline resistance alleles was formed in premeiotic, mitotically dividing germline cells, which then gave rise to multiple gametes sharing the same resistance allele.”

- Ref. 17 states that *“flies from the same mother were significantly more likely to have a sibling with an identical resistance allele”*. In comparison we show that in all 20 cases of maternal gene drive inheritance at the *white* locus (11 for *vasa* and 9 for *nanos*) each F1 female gave rise to only one indel, as found in the analyzed F2 progeny.
- Ref. 17 *“suggests a mechanism in which at least a significant fraction of germline resistance alleles was formed in premeiotic, mitotically dividing germline cells”*. In our work we expand on this by evaluating, in multiple situations, the amount of indels generated due to maternal effect, in the germline of F1 females (by sampling F2

males). Together with the inheritance observed in such cases (**Fig. 2D,E**), this quantification allowed us to pinpoint the timing of conversion to the zygote stage for *white*, and most likely syncytial blastoderm for *yellow*.

Overall, we think that references 2 and 17 brought important information to the gene drive field. Our results using the tGD system allowed us to dig deeper on the maternal effect and clearly distinguish how it impinges on gene drive inheritance in different crosses performed in **Fig 2** (see comparison in **Fig. 2C, D, E**). Importantly, this allowed us to pinpoint the timing of allelic conversion events simultaneously happening at two different loci.

12. Figure 3 C-J: I think the take-home message for these will be lost for most readers. Additionally, the format could be improved. Maybe bars of variable height for different numbers of sequences, with the bars divided based on how many unique sequences there are?

(12) We have explored different graphical representations including the one suggested by Reviewer #1 and concluded that the current scheme is the most graphically informative way to convey our message: the image aims at pointing out the number of different indels recovered in the F2 progeny of each F1 cross (representative of the germline of individual F1 females). We believe this aspect would be harder to appreciate with other graphical representations.

We believe that we properly explained this point in the figure legend in order to help the reader. We thank the reviewer for the suggestion, but we decided to retain the current representation. However, we did follow the reviewer's suggestion on increasing the font size of the numbers to increase legibility.

13. Figure 4A: maybe note that the mosaicism was due to leaky somatic expression from *vasa*-Cas9? This should be clear comparing to *nanos*, which shouldn't have such expression in significant quantity.

(13) We thank the reviewer for the comment, we have discussed this aspect in relation to our model on page 11 lines 227-230 and used to infer the timing of the cutting event.

In our hands, both *vasa* and *nanos* promoters performed similarly for what concerns the eye mosaicism observed in the F1 females when Cas9 and gRNAs came together from the male (**Fig. 4A**).

In previous work (Ref 17), the authors suggest that *nanos* could be a better choice for gene drive purposes since in their hands the *nanos* promoter seems to lack somatic leakiness. We believe that this difference could be due to lower activity of the used *nanos* construct which in this study uses a different Cas9 codon sequence and NLS number (difference highlighted above as a response to comment #11).

14. A general comment at this point: there doesn't seem to be any direct measurements of reporting of the germline and embryo resistance allele formation rates in these construct, even though the necessary data seems to be present.

(14) In our work we decided to focus on analyzing the factors that affect the efficiency of allelic conversion in a gene drive system, rather than expanding on the generation of resistant alleles. We perform an in depth characterization of resistance alleles in the germline of single F1 females by extensively sequencing resistant alleles from F2 males (representative of the F1 germline since F2 males only have on X chromosome) generated during the drive process which we believe sheds some light on the timing of Cas9 activity, and the preferential generation of specific indel alleles.

15. The small molecule control method (the same method from partly the same group) was already introduced in this other manuscript (<https://www.biorxiv.org/content/10.1101/665620v1>). It's fine to include in the authors' manuscript, but it doesn't show anything truly different compared to the previous manuscript, other than a brief test in tGDs.

(15) The small-molecule BiorXiv manuscript that the Reviewer #1 pointed out is a proof-of-concept demonstrating that a gRNA-only split gene drive can be controlled by the addition of a small molecule using the system that we have developed.

In this manuscript we instead show:

- 1) That the small-molecule system can be applied to the tGD which differently has the same invasive potential of a full gene drive, and therefore could be used in future laboratory experiments to test the potential control of the localized spread of gene drives.
- 2) We use the system as a tool to activate Cas9 specifically in the adult F1 female germline and demonstrated, for the first time, that gene drive can be generated in this tissue (**Fig. S6A, C, C'**). In all previously published work, Cas9 is expressed under the control of germline regulatory regions that are expressed as early as the developing embryo rendering hard to tease apart the timing of allelic conversion events. We show that the drug controlled system can be used to restrict the activity to a determined window in development. This is an important new piece of information regarding the mechanisms and timing of the gene drive process.

We believe we have adequately described these points and our reasoning in the manuscript:

"Our above-presented studies on female germline resistance suggest the Cas9-induced cleavage events could happen as early as the zygote stage. As such alleles pose a potential problem to gene drive applications, we wondered how allelic conversion would

perform when solely restricted to the adult germline. No published gene-drive work has thus far been able to precisely establish the timing of drive conversion events.” [...] “Here, we first used a comparable DD2-Cas9 line and showed that mentioned drug-regulated system could be applied to the tGD(y,w) for controlling its super-Mendelian inheritance (Supplementary Fig. 6; Supplementary Data 3). Next, we used the TMP regulation in our tGD system and were able to activate Cas9 only in the adult female germline, showing that super-Mendelian inheritance can be achieved when the gene drive process is restricted to this tissue (Supplementary Fig. 6; Supplementary Data 3), although resistant alleles were also detected (Supplementary Fig. 7).

This approach opens a new avenue for restricting Cas9 activity to a proper window when HDR is favored, perhaps representing a way to bypass the maternal effect”.

16. The concept of impaired homology reducing drive efficiency is not novel or unexpected and was first measured in reference 17 and examined in more detail (with modeling and additional experiments) in the following bioRxiv reference (<https://www.biorxiv.org/content/10.1101/679902v1>). These indicated that impaired homology on even one side of the cut was enough to reduce drive conversion efficiency. The authors may not have seen this due to the small size of their impaired homology. The interesting part of this section is that doubly impaired homology seems to more substantially impair drive conversion than predicted by a multiplicative combination of impaired homology on either side. It is weird that it seemed to affect the Cas9 element too a bit (maybe reduced gRNA expression), but the lack of statistical comparisons makes one pause before drawing meaningful conclusions.

(16) We thank the reviewer for pointing out the Biorxiv manuscript in question, we probably missed it since it was made public only 10 days before our submission of this manuscript; following the suggestion we added this reference in the manuscript.

We believe that our study differs from the study in reference 17 and the mentioned bioRxiv manuscript: while these two studies focus on the use of multiplexing to increase gene drive efficiency, our study looks at the effect of impaired homology on the allelic conversion efficiency.

Here are the differences that we have identified:

1. In the Reference 17, authors used two gRNAs with cut sites 100 bp apart. These gRNAs are driven by different U6 promoters and could be expressed at different levels and/or different timing windows, and having different efficiencies. These aspects add complexity to the system and would make it harder to tease apart exactly the contribution that missing homology in a subset of such events have on the overall conversion efficiency.

In our work we generate a situation in which impaired homology is generated on each or both sides, and analyzed in a system that controls for gRNA activity (same target), for gRNA expression (same U6 promoter), genomic location (same site) and looks specifically at the effect on conversion efficiency, while maintaining the simultaneously-copying Cas9 construct as an internal control.

2. In the biorXiv manuscript, the authors used up to 4 gRNAs to study the effect of gRNA multiplexing on gene drive efficiency. As for Ref. 17, the change in drive efficiency in different situations could be due to impaired homology in a subset of the conversion events: when using two gRNAs a mix of events can be obtained: a) only gRNA1 cuts, b) only gRNA2 cuts or c) both cut, and d) one of such cuts could be repaired by NHEJ before conversion (We have illustrated this in the reply to Reviewer #2 Comment #8). This complexity of repair templates is exacerbated in 3- and 4-gRNA multiplexing systems.

In our work we instead focus on the analysis of impaired homology, and evaluating specifically how impairment on either or both sides affects conversion efficiency while maintaining only one gRNA in the system. We believe our findings could be informative in the design of future multiplexing drive strategies.

We added the relevant statistical analysis in the **Supplementary Data 4** suggested by Reviewer #1 to evaluate potential differences, and modified the text accordingly.

17. Page 15, lines 372-374: would this provide any benefit over simply using two complete gene drive alleles located at different loci, each with the payload? Since the proposed system has two components and would thus be nearly as difficult to construct as two separate drives, the authors need to show superiority to this method if they want to highlight their tGD as being an interesting alternative.

(17) We believe that there would be advantages of using the tGD over two separate full gene drive constructs for the following reasons:

1. In some of our unpublished data we have observed recombination between similar sequences of different drive constructs that can result in chromosomal rearrangements. This could happen in the likely scenario in which the two full-gene drives would share any sequence similarity (Cas9 coding sequence and its regulatory sequence, U6 promoter, gRNA scaffold). In a tGD system there would be no such sequence similarity.
2. Each construct to be generated would be smaller, leading to easier cloning and transgenesis. If one or multiple effector cassettes would need to be added for the drive implementation, limiting construct size could be beneficial. Additionally, the size of the construct could affect efficiency as the Reviewer #1 points out in a previous comment, although this needs to be investigated further.

3. Lastly, the tGD allows for a different genetic option for deployment. Specific field situations might be best addressed by the use of two full gene drives, while others using a tGD system. The two elements of a tGD are interdependent, and if one does not successfully copy in one individual, the system would stop its propagation in part of the offspring of the given animal. This feature could be used, for example, to spread two effector elements that act synergistically to dampen a mosquito-borne agent.

18. Methods: unfortunately, the methods for plasmid construction are unacceptable. While it is okay to promise full sequences on NCBI at a future date, this makes it impossible to review. Furthermore, the manuscript in any case would need primer sequences and plasmid construction details, plus substantially more information on “standard molecular biology techniques.”

(18) We thank the reviewer for this comment. We have added to the Supplementary Information file a Supplementary Methods section outlining the details of plasmid construction and the primers used to generate our constructs. Additionally, we have deposited to NCBI the full sequence of the constructs used in this work. The accession numbers are provided in the Supplementary Methods section as well, although they have not yet been published by the GenBank Submissions Staff.

19. Methods: the specific code for the MGD_{DrivE} simulations should be made available.

(19) The code for the simulations is available, we added a paragraph in the online methods:

“The code for running the simulation is freely available from the MGD_{DrivE} GitHub repository (<https://github.com/MarshallLab/MGDDrivE>), and the package can be installed on R through CRAN (<https://cran.r-project.org/web/packages/MGDDrivE/>). The inheritance cubes used in these simulations are the “cubeTGD” and “cubeTGD_X” variants of the codebase”.

Minor issues:

20. Abstract: “bypassing the dictates of Mendelian genetics” is clear for people who understand gene drive, but it won’t help people who are new to the field learn what it is.

(20) We thank the reviewer for this comment, we realize that the first sentence of the abstract might not have been clear for a more general audience. We changed it from:

“CRISPR-based gene drives spread through populations bypassing the dictates of Mendelian genetics, offering a population-engineering tool for tackling vector-borne diseases, managing crop pests, and helping island conservation efforts”

To:

“CRISPR-based gene drives are capable of spreading through wild populations by biasing their own transmission beyond the 50% value predicted by Mendelian inheritance. These technologies offer population-engineering tools for combating vector-borne diseases, managing crop pests, and supporting island conservation efforts”

21. Introduction, paragraph 2: I don't think its accurate to say that only two approaches are “feasible”. After all, the authors' tGD approach was certain “feasible” before they developed their constructs.

(21) We changed “are feasible” to “have been experimentally evaluated”

22. Page 6, line 125: “alleviating” isn't an accurate word to use here (or as used elsewhere in the manuscript). I'm not exactly sure what the authors are trying to convey.

(22) We changed this part to better explain our reasoning.

23. Page 6, line 128: w118 should be w1118.

(23) We corrected this mistake.

24. Page 7, line 145-146: cite the other studies or refer to only one study.

(24) We added additional references.

25. Page 7, line 148: pathway should be pathways, though this phrase is a bit awkward in the first place. Maybe delete the word “one”?

(25) We corrected this mistake.

26. Page 8, line 152: references 2 and 17 specifically showed this.

(26) We added the suggested references.

27. Page 8, line 156: “in” should perhaps be “is”.

(27) We corrected this mistake.

28. Page 8, lines 185-190: it is unclear if the authors are looking at resistance alleles to analyze resistance alleles formed from maternal effects or resistance alleles formed in the germline of their F1 females. It appears to be the latter, but the paragraphs starts out discussing the former.

(28) We are analysing F2 males resistant alleles as a proxy for the output of the F1 Females germline, which is the aspect analyzed throughout the manuscript. We believe that we clarified this point in the comment #11 above.

29. Page 9, line 201: what is quality here?

(29) We modified the text from:

“...and observed that the ratios of recovered resistant alleles and their quality can vary drastically”

To:

“...and observed that the ratios and the type of recurrent alleles recovered can vary drastically between different drive configurations”

30. Page 9, line 204: If it looks like four it could be a lot more than four. There are only four types of nucleotides, making it appear that this is the max...

(30) In the manuscript we state that in our analysis “we detected a range of 1–4 different resistant alleles per vial”. With this statement we refer to the fact that we have identified 1-4 different indel alleles (different DNA sequences) in each vial (F2 progeny) which is representative of the output of a single F1 female germline.

31. Figure 5A: maybe include the drive without impaired homology for comparison? It would be repeating previously shown data, but the comparison could be worth it. Alternatively, just indicate what this level is.

(31) We thank the reviewer for this comment, we have debated on this point at length as we too believe that it would be beneficial to have the non-impaired homology data for comparison in Fig. 5. As we cannot repeat the same data in two figures within the manuscript, upon the reviewer’s suggestion, we have added in Fig. 5 two marks on the Y-axis highlighting the average inheritance values observed when the transgene presented perfect homology (Fig. 2A, E).

32. Page 16, lines 378-380: this is presumably because the two components will be together more often despite imperfect conversion rates.

(32) We have modified the text by adding the Reviewer’s observation.

33. Supplementary Figure 1: the term “short HAs” is not a very good description of the homology arms here. It would conjure an image of a HA on a plasmid that is just shorter than the usual ~1kB. Instead, the authors may want to use “truncated HA” or something similar the implies that the HAs do not extend to the cut site.

(33) We followed Reviewer #1 suggestion and modified the nomenclature from “Short” to “Truncated” throughout text and Figures (Fig. 5 and Supplementary Fig. 1).

Overall, the authors should substantially revise their manuscript to incorporate previous work into their discussion and clarify several issues. Much of the manuscript can be presented as useful confirmatory results, but the manuscript does need to have many of the above revisions implemented to be scientifically sound and ready for publication.

(34) We are grateful for Reviewer# 1’s assessment which helped to improve the manuscript. We have addressed all the concerns raised to the best of our abilities and we are providing a revised version of the manuscript including the suggested edits and revisions.

Reviewer #2 (Remarks to the Author):

This manuscript addresses several issues that are central to gene drive development and potential deployment including safe propagation, spatially/temporally restricted activity, resistant allele generation and use of multiple gRNAs. In general the experiments are clearly presented and the text is easy to follow. The figures are also well constructed although the text in some sections is very small and is illegible on a standard print version.

The (trans) drive design has not been published previously and is an interesting concept – a significant advantage of this approach is that the drive components can be individually propagated in complete safety. Further, multiple CAS9 promoters and gRNA targets can be readily combined and compared side-by-side for allele conversion and indel-generating activity. Additionally, as far as I am aware, this is the first demonstration of multiple gene conversion events in a single individual.

Although not explicitly mentioned, the data have implications for “Daisy Drive” strategies which are predicated on multiple “simultaneous” homing events within an individual.

(1) We thank Reviewer #2 for this observation, we have added a comment about the potential implications of our work using two simultaneously-copying genetic elements for daisy-chain drives or other systems using two or more genetic elements.

“Furthermore, these findings have implications for other strategies that similarly use multiple elements driving simultaneously such as the proposed “daisy-chain drive”²⁴ or integral gene drives²⁵ .”

Comments

Fig. 1, 2 & 3: Thorough demonstration of the concept. One difference between Fig 1A and 2A that is not discussed is that the homing events in the latter occur on the same chromosome which may impact homing efficiency. Notably this is higher in 2A (same chromosome) than 1A (different chromosome). Can the authors speculate about this? It would be interesting to also generate the yellow/ebony drive (Fig. 1) using Nanos promoter – this would also help to address test whether allele conversion on the same chromosome is more efficient. Also, simultaneous cutting could result in chromosome translocation or inversion – was this investigated?

(2) We thank Reviewer #2 for the comment, we have added a note on the difference highlighted by Reviewer #2 and discussed it in the text:

*“Interestingly, the same Cas9-Red transgene displayed a significantly higher inheritance rate in the tGD(y,w) (89%) than in the tGD(y,e) (83%) (statistical analysis in **Supplementary***

Data 2); this difference might result from positional effects modulating the *y1*-gRNA expression when inserted at a different genomic location (*white* or *ebony*, respectively), or perhaps reflect the distance of the transgenes in the two systems: *tGD(y,w)* are on the same chromosome while *tGD(y,e)* are on different chromosomes.”

- (3) We thank the reviewer for the suggestion, we have performed the *nanos tGD(y,e)* experiment and reported it in the Supplementary Data 1. We have not seen a major difference from our *tGD(y,e) vasa* data suggesting that, in our system, allele conversion at these two loci seems to be consistently lower compared to the *tGD(y,w)*.
- (4) The potential of genomic rearrangements while driving multiple elements is an interesting question, and although in this work it was not investigated, this is an aspect of concern to us as we have observed it in other unpublished systems and we plan to investigate this in depth in future work.

One experiment that I would really like to see is how the *tGD* system behaves in the male germline. The targeting of *yellow* in all experiments precludes this because this locus is X linked. As shown in the modeling, the *tGD* works best when homing occurs in both sexes – however, only females are tested in the manuscript. Therefore I believe that it is also important to test the male germline.

- (5) Upon suggestion of Reviewer #2 we performed the experiment testing *tGD(y,e)* in the male germline and added to the Supplementary Data 1. Since the genetic situation would not be fully comparable to the *tGD* (only the element in *ebony* will undergo gene drive), we maintained this experiment as a minor point and modified the text accordingly.

Is the *nanos* promoter Cas9 construct smaller than *Vasa-Cas9*? This may also impact the efficiency of allele conversion.

- (6) The size of the inserted *vasa-Cas9* construct is 8.3 Kbp while *nanos* is 7.6 Kbp.

Based on our swapped version *tGD(y,w)* vs. *tGD(w,y)* in which we look at copying efficiency of the 3 Kbp gRNA construct and the 8.3 Kbp Cas9 construct at both the *yellow* and *white* loci, we observe that:

- 1) When analyzing our swapped construct data (**Supplementary figure 2**), it seems that at the *yellow* locus the smaller gRNA construct gets copied more efficiently than the Cas9 element: 95% in *tGD(w,y)* vs 89% in *tGD(y,w)* (p-value < 0.0004).
- 2) When we instead look at the *white* locus we see that the larger Cas9 in the swapped *tGD(w,y)* construct (**Supplementary Figure. 2**) is copied also at a significantly higher rate than the smaller gRNA one in our *tGD(y,w)* arrangement (**Fig.2A,E**): 98% in *tGD(w,y)* vs

96% in tGD(y,w) (p-value = 0.038). This suggests that, at least in this case, an increase in construct size seems to correlate with increased efficiency.

Therefore we did not want to make a strong statement regarding transgene size as our data seems inconclusive on this aspect.

Fig. 3./Supp Fig 4: It appears that no F2 progeny with WT sequence were identified from the sequencing analysis – please confirm.

(7) The Reviewer is correct, we have observed 100% cutting in the molecularly characterized flies, 225 flies for *yellow* and 242 flies for *white*. In the cases in which we isolated *w+* male animals they all displayed an in-frame indel that reconstituted function of the protein.

Model in Fig 4 is plausible.

Figure 5. I found this experiment confusing. I disagree with the statement:

“However, as it is intrinsically impossible to have perfect homology to all DNA ends generated when using multiple gRNAs,...”. Cleavage of the WT chromosome at flanking sites will still result in perfect homology to the gene drive bearing chromosome – although the homology will not “but up” against the insertion. As I understand it, the experiment does not examine this scenario – rather it tests the impact of a lack of homology flanking the gene drive sequence. I cant see how these experiments relate to the multi-gRNA scenario.

(8) We thank the reviewer for the comment and we realized that our explanation on how this experiment relates to the gRNA multiplexing required clarification.

By using the sentence “*However, as it is intrinsically impossible to have perfect homology to all DNA ends generated when using multiple gRNAs,...*” we meant that when using multiple gRNAs the system is no longer binary anymore (cut vs. no cut). When using two gRNAs, for example, there are potential scenarios in which one cut is generated at a different time than the second. In this situation a non-homologous overhang is generated. Another situation in which this happens is when one cut happens on one side, and gets repaired by NHEJ generating an indel (green * in the figure below), and the second cut happens after repair has gone to completion. We have illustrated in a graphic below all the possible repair templates generated in a 2-gRNA scenario (left) and events resulting in a double-overhang when using 3 (or more) gRNAs (right).

We understand that our experiment does not exactly reflect the situation of a multiplexing gene drive as it is using only one gRNA, nonetheless our experimental design allowed us to explore how HDR repair is affected in situations approximating some of the cut events of a multiplexing gene drive.

We have modified (**bold underlined**) the paragraph to more explicitly convey our message:

*“However, it is intrinsically impossible to have perfect homology to all **possible** DNA ends generated when using multiple gRNAs. **For example, when using two gRNAs, one cut can be generated earlier and repaired by NHEJ generating an indel at that location. Subsequent cutting by the second gRNA would generate a repair template carrying a non-homologous overhang on one side and perfect homology on the other side.** We wondered to what extent this **potential** homology discordance between the cleaved chromosome and the allele to be propagated would affect the efficiency of a gene drive.”*

*(...) **“For what concerns gene drive our data suggests that multiplexing would be most efficient when using two gRNAs, with PAMs pointing towards each other (PAM-in) generating cuts within the tested range of about 20 bp. Increasing the distance between the two cleavage sites would generate longer overhangs could result in altered efficiency. Using three or more gRNAs would, in certain situations, generate repair templates with two, non-homologous overhangs that dramatically lower HDR-mediated repair.”***

Fig. 6: Preliminary modeling is included – useful addition to the manuscript. Some modeling analysis of how the trans system behaves for population suppression (rather than replacement) would be valuable.

(9) We thank the Reviewer #2 for this suggestion which helped to improve the manuscript. We also modeled the tGD behavior comparing it to the full gene drive in a specific population suppression scenario. Consequently, our manuscript contains an additional figure which we placed in the Supplementary Material as **Supplementary Figure 9**.

Additionally, we discussed these new results and modified the text accordingly at the end of our results and discussion section:

“To further explore the potential of the tGD in the field, preliminary modeling of our system intended for population suppression was studied. In this case, the gRNA locus targets a gene required in at least one copy for female fertility, in this situation the tGD system behaves analogously to an equivalent full drive approach intended for population suppression but we

envisioned some potential benefits in the tGD arrangement compared to the full-GD (Supplementary Fig. 9). For an allelic conversion frequency of 100%, both the full-GD and tGD systems induce a population crash within 1.5 years of the releases; however the population quickly rebounds in both cases for an allelic conversion frequency of 99% due to resistant alleles emerging at the female fertility locus that preserve fertility, thus conferring a selective advantage and preventing a crash (Supplementary Fig. 9). Tolerable rates of resistant allele generation are related to the inverse of the population size that one wishes to suppress³³. Interestingly, one potential advantage of the tGD population suppression system is that a disease-refractory cassette could be linked to the Cas locus such that, in the event of resistant allele generation at the gRNA locus, the disease-refractory trait would still be propagated (Supplementary Fig. 9)”.

Minor issues

“...alleviating the significant difference between the two transgenes that was observed for tGD(y,w) (89% at yellow and 96% at white)...” Significant difference implies a statistical analysis was performed. This does not appear to be the case.

(10) We have added the statistical analysis as Supplementary Data 2.

Fig 3. The “Flies sequenced” numbers are very small and difficult to read.

(11) We have increased the font size by a point to make the figure more legible.

Fig. 4B correct “zigote” (zygote)

(12) We have corrected the typo.

Supp Fig 6 B’ Cas9 versus DD2-Cas9 experiments should be labeled.

(13) We have added the missing labels to Supp. Fig 6.

Reviewers' Comments:

Reviewer #1:

Remarks to the Author:

The authors have overall made a very good effort in their revisions and considerably improved the manuscript. However, I think there are still several very important points that may need additional consideration. Below, I'm using the same numbering system as my original review, and replying to the authors' responses (points 1-2, 7-9, 11, and 16-17 are the main ones for the new revision). I didn't notice any other issues, except that the resolution of the figures should probably be improved.

1. I certainly agree that tGDs are safer than regular gene drives, but I don't think the difference is enough to make this a substantial point of the manuscript. Referring to the authors table, I think the number of flies and the methods involved with handling them mean that the last row would have much higher "weight" than the previous rows, thus making the "overall" risk of the drives comparable.

For example, let's say you keep a gene drive for a year to do experiments. Two distinct lines for one complete gene drive construct with new vials roughly two weeks (probably keeping the old vials as backups, but they'd just be sitting there without any attention, so let's discount them), so that's about 50 vials per year. I don't think you'd really want to expand these lines that much for any experiments, but let's add an additional 10 vials for that. Now comes the experiments. Let's say that you cross flies from each of your expanded bottles for the first generation, so that's 10 vials where the offspring of the tGD are now driving, so these ten counts for both the complete gene drive and the tGD. Then, there are the individual crosses derived from these. In Figure 2, the authors did 156 crosses. This is probably more than really needed (see below, but kudos to the authors on their diligence), so let's go with 100. We now have 160 dangerous vials for the complete gene drive and 110 for the tGD. The tGD is a little better, but this is marginal. The tGD danger window is certainly narrower in time, but I'd argue that other factors making it more dangerous are more important. After all, there is a greater chance of escape from a vial that gets extensive phenotyping and future crosses than one just undergoing normal maintenance. Furthermore, if the gene drive is highly successful, the experimenters would want to do a cage trial. Cage trials result in far more escapees than individual crosses, and they also use many more flies. Such cage trials would be "dangerous" equally for the complete gene drive and the tGD. Thus, the actual improvement in safety is relatively small for the tGD unless very few experiments are done and the line is kept for a long time.

Regarding the ease of testing multiple combinations of gRNA and promoter elements, this is certainly true compared to full gene drives. However, split-drive (where the Cas9 element doesn't have an associated gRNA for HDR) systems are similar in this regard and these systems also provide a better degree of biosafety, so it's less clear what the utility of tGD is compared to split systems.

2. Bullet point 1: split drive systems have already been shown in yeast and flies where one element helps drive another. This isn't even specifically needed for Daisy drives (or integral drives) to be considered technically feasible, and the manuscript does not directly demonstrate a Daisy drive. Thus, I'm not sure if this manuscript contributes to showing that Daisy drives are feasible in principle (the issue with Daisy drives seems to be technical rather than conceptual). The idea of using a gRNA to drive the Cas9 component is novel, and the experimental demonstration of using a gRNA to drive another element is also novel. However, I don't think the authors have made the case that these novel findings are very impactful for future implications, or that their experimental systems work differently than would be predicted based on previous knowledge, or that it requires construction methods that are different than predicted. Bullet point 2: Yes, though as described above, split drives would be a way to test combinations rapidly and with a lot more safety. The idea of a tGD release candidate being amplified separately has unclear advantages. If there is already permission for a release, why would a

small number of premature escapees matter much for biosafety? On the bright side, it would prevent potential confusion when studying the system after release (assuming that the escapees did well), which is a definite advantage of the system. However, unless the tGD elements are highly efficient, they would separate at a noticeable rate, somewhat degrading drive performance. This may outweigh the goal of preventing slightly premature accidental release of driving insects. Bullet point 3: See point 17 below.

3. I'm happy with this revision.

4. I am happy with the expansion of the introduction, with caveats to some points as described above in #1 and #2.

5. It's not a big deal if the authors felt that the section on ebony should stay in the main manuscript. I think the authors for assessing male drive in ebony. While not a tGD, it still helps the field overall a bit to get more comparisons of male and female driving efficiency at different genomic loci, even if there wouldn't be anything unexpected (at least nothing predicted... maybe we just need more sites before something weird and interesting pops up).

6. I'm happy with this revision, but maybe bring a few of the most important p-values into the main text, just to make it clear that the differences are real for people who don't go into the supplement.

7. If location of expression and target makes a difference, then Figure 2 and Supplementary Figure 2 are still the comparisons of importance. The potential difference from size is larger in Figure 2 than Supplementary Figure 2. Thus, the conclusion could be that size matters, but location effect was enough to overcome this in the Supplementary Figure 2 drive. A statistical model with effect from both location and size could potentially demonstrate this. I don't consider this a critical point, though, just an opportunity for the authors to potentially squeeze a little more out of their data.

8. This change is fine. It might be better to make some further clarifications, though. Consider changing

"indicating no maternal inheritance effect is observed when Cas9 is inherited by the mother in the tGD arrangement"

to:

"indicating no reduction in the inheritance rate when maternal Cas9 but not gRNA is received from the mother in the tGD arrangement"

9. The fact that both Cas9 and gRNA must be maternally deposited to make resistance alleles was shown in reference 21, which directly translates to drive inheritance reduction, as shown in many studies. Thus, reference 21 should still be cited for this conclusion, rather than just the conclusion that Cas9 is not sufficient alone to reduce drive inheritance. Measurement of average drive inheritance when both Cas9 and gRNAs are deposited by the same parent is just another way to show this. This just corresponds to the complete drive, after all. I'm not saying that the authors' did anything wrong in their experiment or conclusion, just that the conclusion was shown in a slightly different way in the earlier study, and that this should be cited. It could be a matter of simply changing:

"suggesting that a strong maternal effect on a gene drive is generated only when the two elements are inherited together from a female germline."

to:

"suggesting that a strong maternal effect on a gene drive is generated only when the two elements are inherited together from a female germline, consistent with previous results (21)."

10. I'm happy with this response.

11. Batches were not important for reference 2 since they merely reduce the statistical power. Reference 17 did use single individuals. However, it looks like the authors may have been confused by my point here. These sequenced alleles that were discussed were NOT formed by maternal deposition. Maternal deposition was considered elsewhere in these references.

Referring to maternal deposition, reference 2 and 17 still found the same thing by direct sequencing of females with the drive and with the yellow or white phenotypes. In the former, sequences were mosaic, including multiple distinct sequences. In the latter, there was a single sequence (compared in reference 17). This exactly corresponds to the conclusions in this manuscript and should be referenced. The current manuscript simply looked at progeny. This wouldn't be expected to provide any new information for the white, but it does show that for yellow, different sequences were also within the germline (and gametes), in addition to the whole individual (which could have had the same sequence in the germline, even if different sequences were present in other regions of the body). Overall, the authors should cite the previous references and then carefully show where they extended them.

12. Not a big deal. The increased font size does help a bit, which is useful.

13. Interesting. It might be worth mentioning that the nanos promoter in reference 17 did not produce any mosaicism when inherited through the male line, in contrast to the results from the author's nanos drive. It could be the Cas9 sequence itself or perhaps the genetic background?

14. Not a big deal, though would still be nice to have in the supplement.

15. I'm happy with this response.

16. The automatic alert system of bioRxiv does indeed seem to miss relevant publications a fair amount of time. The authors are correct that reference 17 did not explicitly show this in as well-controlled a manner, but it was still strongly suggested by the data and explicitly discussed, making it worthy of citation here. The bioRxiv reference use multiple gRNAs, but it also used a 1-gRNA drive with impaired homology at one end, as done by the authors of the current manuscript. This 1-gRNA drive has a greater impairment and showed a notable effect. Again, the difference is that current manuscript tested a drive that had impaired homology on both sides. All of this should be discussed. Finally, the suggestion to use two good ones may be suitable for maximizing drive inheritance, but there is still the difference between types of resistance alleles to consider.

17. Such unpublished data should be included in the manuscript if it supports an important point, or the authors should at least suggest the possibility of two drives having such events (and the tGD being resistant to them - Cas9 and gRNA scaffolds can and have been easily recoded, but the point with the other regulatory elements stands). On the other hand, wouldn't a chromosomal rearrangement be quickly removed from the population when the rearranged chromosomes were not inherited together? Regarding size, yes, this could be a real advantage too, and could perhaps be discussed. On the other hand, reduced gRNA activity due to only one Cas9 in tGD (compared to two in two complete drives) may reduce performance. Regarding point 3, I am unclear on the benefits. If two effectors are interdependent, then wouldn't the possibility of separation in a tGD substantially reduce overall efficiency? It's not immediately apparent to me when there would be an actual benefit to tGD vs. two gene drives, except in matters of efficiency noted above.

Overall, these considerations are worth discussion, but they don't directly address my main point that a tGD should definitely be compared to two complete gene drives and not one in the actual modeling.

It will likely show slightly worse performance than two complete drives, but that's not a big deal. One possibility is to do this comparison, but then have the above discussion above about why tGD may actually be better than two complete drives, even if it has similar performance in the model that doesn't take into account these possibilities that are not included in the simple model.

18. I am happy with this revision.

19. I am happy with this revision.

20-33. I am happy with all of these this revisions and responses.

Extra note on 31: I think it's fine to repeat data in a manuscript when appropriate. The solution implemented by the authors here is probably better for this scenario, though. If I may make an aesthetic suggestion, it would be nice if the red and green lines crossed the black axis line, to emphasize that they come from data (inside) and make them a little easier to see.

Reviewer #2:

Remarks to the Author:

Thank you for your comprehensive response to the issues that I raised. The additional experiments and modifications to the manuscript have satisfied all my concerns.

We are grateful that both Reviewers' appreciated our effort to improve the manuscript. Here, we provide a revised manuscript addressing additional comments of Reviewer #1 in blue.

Reviewers' comments:

Reviewer #1 (Remarks to the Author):

The authors have overall made a very good effort in their revisions and considerably improved the manuscript. However, I think there are still several very important points that may need additional consideration. Below, I'm using the same numbering system as my original review, and replying to the authors' responses (points 1-2, 7-9, 11, and 16-17 are the main ones for the new revision). I didn't notice any other issues, except that the resolution of the figures should probably be improved.

1. I certainly agree that tGDs are safer than regular gene drives, but I don't think the difference is enough to make this a substantial point of the manuscript. Referring to the authors table, I think the number of flies and the methods involved with handling them mean that the last row would have much higher "weight" than the previous rows, thus making the "overall" risk of the drives comparable.

For example, let's say you keep a gene drive for a year to do experiments. Two distinct lines for one complete gene drive construct with new vials roughly two weeks (probably keeping the old vials as backups, but they'd just be sitting there without any attention, so let's discount them), so that's about 50 vials per year. I don't think you'd really want to expand these lines that much for any experiments, but let's add an additional 10 vials for that. Now comes the experiments. Let's say that you cross flies from each of your expanded bottles for the first generation, so that's 10 vials where the offspring of the tGD are now driving, so these ten counts for both the complete gene drive and the tGD. Then, there are the individual crosses derived from these. In Figure 2, the authors did 156 crosses. This is probably more than really needed (see below, but kudos to the authors on their diligence), so let's go with 100. We now have 160 dangerous vials for the complete gene drive and 110 for the tGD. The tGD is a little better, but this is marginal. The tGD danger window is certainly narrower in time, but I'd argue that other factors making it more dangerous are more important. After all, there is a greater chance of escape from a vial that gets extensive phenotyping and future crosses than one just undergoing normal maintenance. Furthermore, if the gene drive is highly successful, the experimenters would want to do a cage trial. Cage trials result in far more escapees than individual crosses, and they also use many more flies. Such cage trials would be "dangerous" equally for the complete gene drive and the tGD. Thus, the actual improvement in safety is relatively small for the tGD unless very few experiments are done and the line is kept for a long time.

Regarding the ease of testing multiple combinations of gRNA and promoter elements, this is certainly true compared to full gene drives. However, split-drive (where the Cas9 element doesn't have an associated gRNA for HDR) systems are similar in this regard and these

systems also provide a better degree of biosafety, so it's less clear what the utility of tGD is compared to split systems.

The original issue raised by the reviewer seemed to concern our improper use of the word "safe" to describe our system as the tGD has the similar spread potential of a full Gene Drive. As we mentioned previously we have revised the manuscript modifying the title and advertising the technology as "safer" than a full gene drive, and we appreciate that the reviewer is in agreement with this point.

As we understand from the comment above, the reviewer is now raising a new point concerning how much safer than a full gene drive a tGD really is?

Concerning the laboratory maintenance and experiment outlined in the reviewer's example, it seems to us that the conclusion is in agreement with our claim that the tGD system is at least marginally safer during experimental use and certainly is much safer for long term storage of stocks.

We believe that an additional important safety feature would be in preparation for potential field releases in which, in the case of a full GD, a single full gene drive fly/mosquito if escaped before the scheduled release could disseminate the drive element in the local population whereas a single escape of a fly/mosquito carrying only one of the two tGD components would not be able to do so. (We expand on this point on our reply to Comment #2)

In the introduction we clearly state that the tGD system combines the strengths of both full gene drives and gRNA-only drives. The tGD has the potential of being tested in the lab and applied in the field displaying the same characteristics as of a full gene drive, while maintaining the safer property of a gRNA-only drive where Cas9 and gRNAs can be maintained. We believe we have addressed this in the manuscript with these sentences:

"The use of a full-GD is causing concern to the scientific community as an accidental release could spread unchecked"¹⁵. While a gRNA-GD would address such concerns, its application in the field for large scale population engineering is unlikely to succeed since it would require that a large percentage of the population carried a Cas9 transgene²² "

2. Bullet point 1: split drive systems have already been shown in yeast and flies where one element helps drive another. This isn't even specifically needed for Daisy drives (or integral drives) to be considered technically feasible, and the manuscript does not directly demonstrate a Daisy drive. Thus, I'm not sure if this manuscript contributes to showing that Daisy drives are feasible in principle (the issue with Daisy drives seems to be technical rather than conceptual). The idea of using a gRNA to drive the Cas9 component is novel, and the experimental demonstration of using a gRNA to drive another element is also novel. However, I don't think the authors have made the case that these novel findings are very impactful for future implications, or that their experimental systems work differently than would be predicted based

on previous knowledge, or that it requires construction methods that are different than predicted. Bullet point 2: Yes, though as described above, split drives would be a way to test combinations rapidly and with a lot more safety. The idea of a tGD release candidate being amplified separately has unclear advantages. If there is already permission for a release, why would a small number of premature escapees matter much for biosafety? On the bright side, it would prevent potential confusion when studying the system after release (assuming that the escapees did well), which is a definite advantage of the system. However, unless the tGD elements are highly efficient, they would separate at a noticeable rate, somewhat degrading drive performance. This may outweigh the goal of preventing slightly premature accidental release of driving insects. Bullet point 3: See point 17 below.

- We appreciate the reviewer's comment about the novelty of our system. We previously followed Reviewer #2 suggestion to comment on the implications that our data might have for technologies using multiple elements, and state that:

“...these findings have implications for other strategies that similarly use multiple elements driving simultaneously such as the proposed “daisy-chain drive” or integral gene drives.”

We do not believe that our sentence claims that our data directly demonstrates daisy or integral drive. We do instead believe that our data, in which two elements are simultaneously copying, is encouraging for the future development of strategies needing to drive multiple elements.

- There are two types of field trials: 1) enclosed field release, and 2) open field release. Often (1) would precede (2). In enclosed field releases genetically modified individuals (let us say mosquitos) are amplified *in loco* and released in a population derived from the local population, usually caged outdoors. This kind of trial would aim at no escapees. This is usually achieved by double containment and strict monitoring of traps around the experimental site. In the setup process, gene drive individuals need to be amplified in numbers before the release happens. Assuming few generations of amplification the gene drive mosquitoes would be present for an additional few months before the release. If using a tGD instead of a full-GD any escapee from either the gRNA or the Cas9 cage would be incapable of spreading in the wild, contrary to a full gene drive.

3. I'm happy with this revision.

4. I am happy with the expansion of the introduction, with caveats to some points as described above in #1 and #2.

5. It's not a big deal if the authors felt that the section on ebony should stay in the main manuscript. I think the authors for assessing male drive in ebony. While not a tGD, it still helps the field overall a bit to get more comparisons of male and female driving efficiency at different

genomic loci, even if there wouldn't be anything unexpected (at least nothing predicted... maybe we just need more sites before something weird and interesting pops up).

6. I'm happy with this revision, but maybe bring a few of the most important p-values into the main text, just to make it clear that the differences are real for people who don't go into the supplement.

7. If location of expression and target makes a difference, then Figure 2 and Supplementary Figure 2 are still the comparisons of importance. The potential difference from size is larger in Figure 2 than Supplementary Figure 2. Thus, the conclusion could be that size matters, but location effect was enough to overcome this in the Supplementary Figure 2 drive. A statistical model with effect from both location and size could potentially demonstrate this. I don't consider this a critical point, though, just an opportunity for the authors to potentially squeeze a little more out of their data.

We thank the Reviewer for this advice. We believe the observed effects have been already addressed and discussed in the manuscript following the Reviewer's suggestion. While we believe that the size could impact efficiency, we do not believe we can make conclusions on this aspect from our data.

8. This change is fine. It might be better to make some further clarifications, though. Consider changing

"indicating no maternal inheritance effect is observed when Cas9 is inherited by the mother in the tGD arrangement"

to:

"indicating no reduction in the inheritance rate when maternal Cas9 but not gRNA is received from the mother in the tGD arrangement"

We modified the text accordingly.

9. The fact that both Cas9 and gRNA must be maternally deposited to make resistance alleles was shown in reference 21, which directly translates to drive inheritance reduction, as shown in many studies. Thus, reference 21 should still be cited for this conclusion, rather than just the conclusion that Cas9 is not sufficient alone to reduce drive inheritance. Measurement of average drive inheritance when both Cas9 and gRNAs are deposited by the same parent is just another way to show this. This just corresponds to the complete drive, after all. I'm not saying that the authors' did anything wrong in their experiment or conclusion, just that the conclusion was shown in a slightly different way in the earlier study, and that this should be cited. It could be a matter of simply changing:

"suggesting that a strong maternal effect on a gene drive is generated only when the two elements are inherited together from a female germline."

to:

"suggesting that a strong maternal effect on a gene drive is generated only when the two elements are inherited together from a female germline, consistent with previous results (21)."

We modified the manuscript and added the citation.

10. I'm happy with this response.

11. Batches were not important for reference 2 since they merely reduce the statistical power. Reference 17 did use single individuals. However, it looks like the authors may have been confused by my point here. These sequenced alleles that were discussed were NOT formed by maternal deposition. Maternal deposition was considered elsewhere in these references.

Referring to maternal deposition, reference 2 and 17 still found the same thing by direct sequencing of females with the drive and with the yellow or white phenotypes. In the former, sequences were mosaic, including multiple distinct sequences. In the latter, there was a single sequence (compared in reference 17). This exactly corresponds to the conclusions in this manuscript and should be referenced. The current manuscript simply looked at progeny. This wouldn't be expected to provide any new information for the white, but it does show that for yellow, different sequences were also within the germline (and gametes), in addition to the whole individual (which could have had the same sequence in the germline, even if different sequences were present in other regions of the body). Overall, the authors should cite the previous references and then carefully show where they extended them.

We addressed Reviewer's concern and modified the text accordingly to discuss how the added references relate to our work.

12. Not a big deal. The increased font size does help a bit, which is useful.

13. Interesting. It might be worth mentioning that the nanos promoter in reference 17 did not produce any mosaicism when inherited through the male line, in contrast to the results from the author's nanos drive. It could be the Cas9 sequence itself or perhaps the genetic background?

We modified the text to discuss these differences and added reference 17.

14. Not a big deal, though would still be nice to have in the supplement.

15. I'm happy with this response.

16. The automatic alert system of bioRxiv does indeed seem to miss relevant publications a fair amount of time. The authors are correct that reference 17 did not explicitly show this in as well-controlled a manner, but it was still strongly suggested by the data and explicitly discussed, making it worthy of citation here. The bioRxiv reference use multiple gRNAs, but it also used a

1-gRNA drive with impaired homology at one end, as done by the authors of the current manuscript. This 1-gRNA drive has a greater impairment and showed a notable effect. Again, the difference is that current manuscript tested a drive that had impaired homology on both sides. All of this should be discussed. Finally, the suggestion to use two good ones may be suitable for maximizing drive inheritance, but there is still the difference between types of resistance alleles to consider.

It seems to us the mentioned bioRxiv manuscript is a work in progress, and there we could not find the information needed to properly compare it to our work, and thus we prefer not to discuss it in detail. Regardless, we have already added both these references at the beginning of our result section in the previous revision.

17. Such unpublished data should be included in the manuscript if it supports an important point, or the authors should at least suggest the possibility of two drives having such events (and the tGD being resistant to them - Cas9 and gRNA scaffolds can and have been easily recoded, but the point with the other regulatory elements stands). On the other hand, wouldn't a chromosomal rearrangement be quickly removed from the population when the rearranged chromosomes were not inherited together? Regarding size, yes, this could be a real advantage too, and could perhaps be discussed. On the other hand, reduced gRNA activity due to only one Cas9 in tGD (compared to two in two complete drives) may reduce performance. Regarding point 3, I am unclear on the benefits. If two effectors are interdependent, then wouldn't the possibility of separation in a tGD substantially reduce overall efficiency? It's not immediately apparent to me when there would be an actual benefit to tGD vs. two gene drives, except in matters of efficiency noted above.

We believe that we already explained our views on the tGD's benefits compared to the use of full gene drives in the previous revision, however we clarify here our reasoning:

- The recombination between similar sequences of different constructs during the drive process is an effect that we have seen in a separate work which is currently being prepared for publication. We did not observe this phenomenon when using the tGD, therefore we do not believe this manuscript is the right place for this data.
- We appreciate that Reviewer #1 agrees with us on the advantage of reduced size of the constructs when using tGD.
- The tGD allows for a different genetic option for deployment that could be taken advantage of in particular situations. In the thought experiment that we described in our previous answer, if one of the two tGD elements does not copy, the offspring of such an individual would carry elements incapable of spreading. If the two effectors carried on the tGD transgenes would act synergistically, they should have an increased chance of being inherited together, only when both elements have been successfully copied. This is an idea that we would like to pursue, and that we have not yet modeled.

Overall, these considerations are worth discussion, but they don't directly address my main point that a tGD should definitely be compared to two complete gene drives and not one in the

actual modeling. It will likely show slightly worse performance than two complete drives, but that's not a big deal. One possibility is to do this comparison, but then have the above discussion above about why tGD may actually be better than two complete drives, even if it has similar performance in the model that doesn't take into account these possibilities that are not included in the simple model.

The original concern brought up by Reviewer #1 on comment #17 asked to provide the reasoning for the benefits of using a tGD instead two separate gene drives, the original comment here below:

“17. Page 15, lines 372-374: would this provide any benefit over simply using two complete gene drive alleles located at different loci, each with the payload? Since the proposed system has two components and would thus be nearly as difficult to construct as two separate drives, the authors need to show superiority to this method if they want to highlight their tGD as being an interesting alternative.”

It seems to us that Reviewer #1 is now asking us to change our comparison to one full gene drive and instead compare it to two gene drives?

First, we are proposing the tGD as an alternative to the use of one, not two full gene drives. Second, we believe that we already explained above the benefits of the use of a tGD would have over the use of two full gene drives:

- Reduce the potential for recombination.
- Reduced size of the constructs.
- Different genetic option for deployment (e.g., safe expansion of separated stocks prior to an intended field release).

We believe the appropriate comparison is the tGD to a single full drive, and not to two gene drives, since this is what the alternative uses would be in practice.

18. I am happy with this revision.

19. I am happy with this revision.

20-33. I am happy with all of these this revisions and responses.

Extra note on 31: I think it's fine to repeat data in a manuscript when appropriate. The solution implemented by the authors here is probably better for this scenario, though. If I may make an aesthetic suggestion, it would be nice if the red and green lines crossed the black axis line, to emphasize that they come from data (inside) and make them a little easier to see.

We thank the Reviewer for this comment, however we believe that the current representation is clear and believe that it should not be a source of confusion as we explained it properly in the figure legend.

Reviewer #2 (Remarks to the Author):

Thank you for your comprehensive response to the issues that I raised. The additional experiments and modifications to the manuscript have satisfied all my concerns.